# Anti-friction gold-based stretchable electronics enabled by interfacial diffusion-induced cohesion

Jie Cao [1], Xusheng Liu[1,2], Jie Qiu [1], Zhifei Yue[1], Yang Li[1], Qian Xu[1,2], Yan Chen[1,2], Jiewen Chen[1], Hongfei Cheng[3], Guozhong Xing [4], Enming Song[5], Ming Wang [1,6] ✉, Qi Liu[1,2,6] ✉ & Ming Liu[1,6]

Stretchable electronics that prevalently adopt chemically inert metals as sensing layers and interconnect wires have enabled high-fidelity signal acquisition for on-skin applications. However, the weak interfacial interaction between inert metals and elastomers limit the tolerance of the device to external friction interferences. Here, we report an interfacial diffusion-induced cohesion strategy that utilizes hydrophilic polyurethane to wet gold (Au) grains and render them wrapped by strong hydrogen bonding, resulting in a high interfacial binding strength of 1017.6 N/m. By further constructing a nanoscale rough configuration of the polyurethane (RPU), the binding strength of Au-RPU device increases to 1243.4 N/m, which is 100 and 4 times higher than that of conventional polydimethylsiloxane and styrene-ethylene-butylene-styrene-based devices, respectively. The stretchable Au-RPU device can remain good electrical conductivity after 1022 frictions at 130 kPa pressure, and reliably record high-fidelity electrophysiological signals. Furthermore, an anti-friction pressure sensor array is constructed based on Au-RPU interconnect wires, demonstrating a superior mechanical durability for concentrated large pressure acquisition. This chemical modification-free approach of interfacial strengthening for chemically inert metal-based stretchable electronics is promising for three-dimensional integration and on-chip interconnection.

Stretchable artificial electronic skins integrated into biological tissues to reliably sense, transmit, and process signals are critical for robust human-machine interfaces (HMIs)[1–4]. An artificial skin is generally composed of two basic components: an electrical functional layer for information acquisition and processing, and a conformal substrate that mechanically matches human tissue/skin and robots[5–7]. These two components are rationally assembled into a functional set using microfabrication, printing or transfer processes, forming various artificial skin devices including stretchable electrodes[8,9], multimodal sensors[10–12], functional circuits[13,14], and standalone stretchable sensing platforms[15,16]. In practical applications, these devices inevitably experience external friction interferences caused by directly

[1]Frontier Institute of Chip and System, State Key Laboratory of Integrated Chips and Systems, Zhangjiang Fudan International Innovation Center, Fudan University, Shanghai 200433, China. [2]School of Microelectronics, Fudan University, Shanghai 200433, China. [3]School of Materials Science and Engineering, Tongji University, Shanghai 201804, China. [4]Key Laboratory of Microelectronic Devices & Integrated Technology, Institute of Microelectronics, University of the Chinese Academy of Sciences, Chinese Academy of Sciences, Beijing 100029, China. [5]Shanghai Frontiers Science Research Base of Intelligent Optoelectronics and Perception, Institute of Optoelectronics, State Key Laboratory of Integrated Chips and Systems, Fudan University, Shanghai 200433, China. [6]Shanghai Qi Zhi Institute, 41th Floor, AI Tower, No. 701 Yunjin Road, Xuhui District, Shanghai 200232, China. ✉e-mail: wang_ming@fudan.edu.cn; qi_liu@fudan.edu.cn

contacting with human skins and fabrics during somatic movement. Friction interferences would give rise to the delamination of the functional layer from the substrate, leading to the deterioration of monitoring signals and even the device failure (Fig. 1a). These issues would be amplified in signal acquisition for vigorous exercise, such as the athlete training and competition that suffers large deformation and rubbing.

In practice, a great number of artificial skins employ chemically inert metals, such as gold (Au) and platinum, as functional layers and interconnect wires to perceive or transmit up-to-date signals, due to their high intrinsic electrical conductivity, good chemical stability and biological non-toxicity[17,18]. For these inert metal-based artificial skins, one of the most challenging issues is the weak adhesion between the inert metals and stretchable substrates due to poor interfacial chemical bonding[19,20], which makes them sensitive to the ubiquitous friction interferences. Although an encapsulation strategy that packages the functional metal layer into endurable polymers has been developed to defense friction interferences, it impedes the metal layer from directly contacting with skins and other media for signal acquisition[21,22]. Another promising strategy is to directly enhance the interfacial adhesion between the metal layer and elastic substrate by

interfacial structural engineering[8,23] and chemical engineering[24]. The former utilizes the configurations of interfacial microstructures such as island-bridge and biphasic interpenetrating structure to compensate the deformation stress imposed on metal layer[8,25,26], but the weak van der Waals forces at the metal-substrate interface results in limited interfacial binding capability. In the later approach, metal layers and elastic substrates are usually chemically modified with specific chemical groups[27–29] or adhesion additives[30,31] to directly strengthen their interfacial binding. However, the chemically modified residues would inevitably penetrate into the inert metal and substrate layers to deteriorate the structural integrity and restrict the electron mobility of the device, resulting in inferior mechanical tolerance and low electrical conductivity.

Here, we report a chemical modification-free diffusion-induced cohesion (DIC) strategy to achieve strong interfacial binding between inert Au and waterborne polyurethane (WPU) for anti-friction electronics. Combining with a rough microporous structure of waterborne polyurethane (RPU), the interfacial binding (peel strength) of the Au-RPU device reaches up to 1243.4 N/m, which is 100 and 4 times higher than that of conventional polydimethylsiloxane (PDMS)- and styrene-ethylene-butylene-styrene (SEBS)-based devices, respectively. Owing

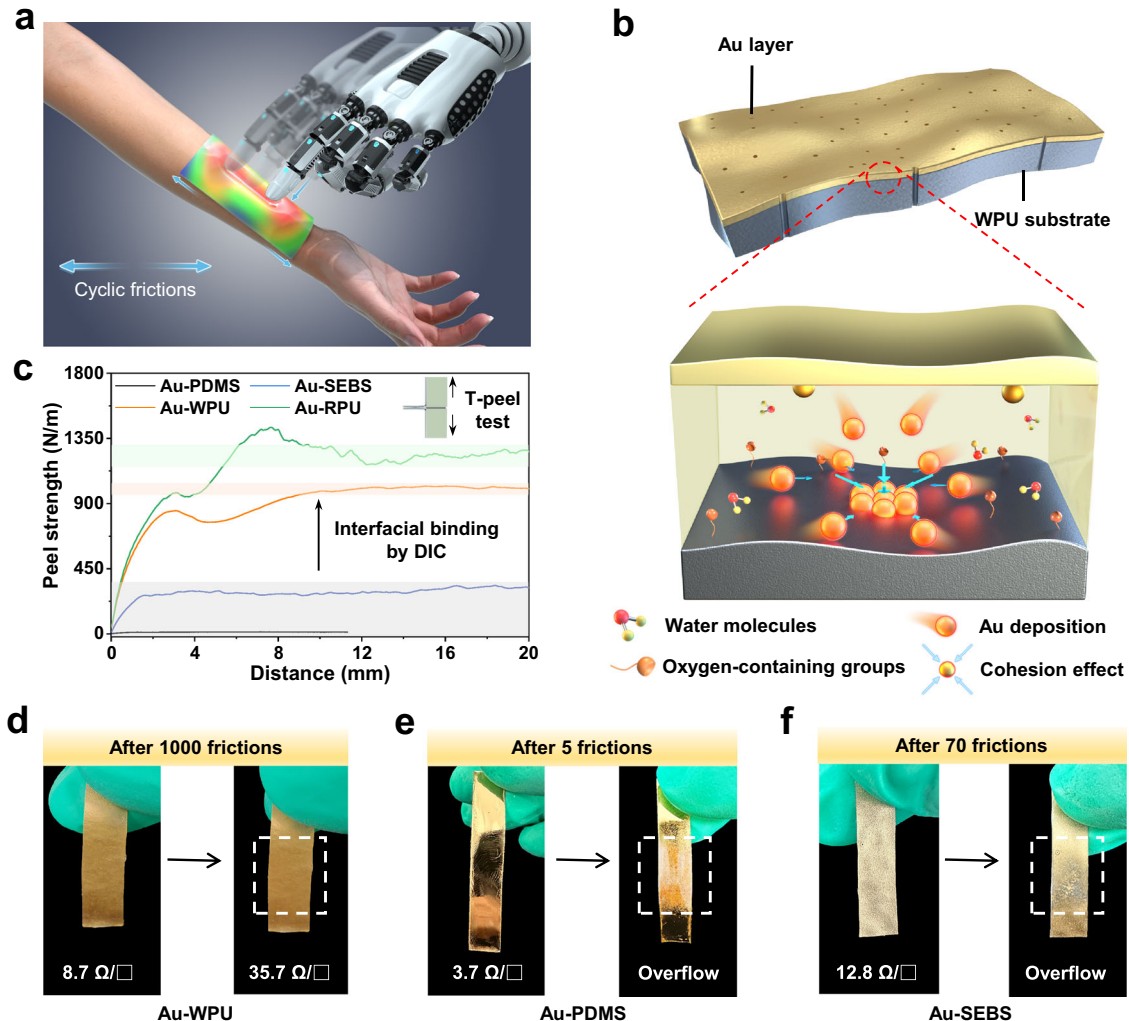

**Fig. 1 | Design of anti-friction artificial skins. a** Schematic illustration of artificial skins suffering from frictions during HMIs applications. **b** Structural illustration of chemically inert Au layer deposited on hydrophilic WPU substrate. Zoom-in view: schematic of interfacial binding between Au layer and WPU substrate through interfacial diffusion-induced cohesion effect. Strong cohesive force is constructed by diffused water molecules and oxygen-containing groups adsorbed on Au grains,

ensuring the strong adhesion of Au grains onto WPU substrate. **c** Comparison of the peel strength. Shaded colors indicate the plateau of the curves. Top right inset shows the schematic of the T-peel test. **d–f** Electrical stability of **d** Au-WPU **e** Au-PDMS and **f** Au-SEBS upon frictions on human forearm with a vertical load. The dashed white area indicates the friction position.

to the strong interfacial binding, the fabricated Au-RPU device can remain highly conductive (electrical sheet resistance <500 Ω/□) even after 1022 ± 76 frictions on artificial skin at a large vertical pressure of 130 kPa, and possesses a low resistivity of $1.09 \times 10^{-3}$ Ω m even when stretched to 400% strain. We further verified the strong interfacial binding is attributed to the cohesion effect of adjacent wetted Au grains in interfacial layer, which is formed by hydrogen bonding of diffused molecular glues (water molecules and oxygen-containing groups on the WPU surface). As a proof-of-concept for anti-friction HMIs, Au-RPU-based stretchable electrodes were fabricated for electrophysiological signal recording even after strong cyclic rubbing by fabrics, hairy scalp and skins, showing a high signal-to-noise ratio (SNR) at 25.0 ± 0.9 dB after 1000 frictions when recording electromyographic signals. In addition, we have developed a stretchable circuit that employs Au-RPU interconnect wires, enabling the illumination of light-emitting diodes (LEDs) even under conditions of frictions and deformation. Furthermore, we have fabricated a pressure sensor array capable of reliably monitoring the concentrated large forces imparted by grasping objects.

## Results

### Fabrication and characterization of anti-friction devices

To achieve anti-friction ability, metallic Au and hydrophilic WPU are considered as the two fundamental components of artificial skin (Fig. 1b). Instead of using conventional elastomers such as PDMS and SEBS, we select hydrophilic WPU as the elastic substrate because of its low Young's modulus, eco-friendly film manufacturing in aqueous solution, high compatibility with other materials and solvents for surface microstructure design, and rich hydrophilic polar groups (Supplementary Figs. 1–3 and Table 1). Such polar groups include oxygen-containing groups and adsorbed water molecules on the surface, which will wet Au grains during Au deposition and form stable aggregates (Fig. 1b and Supplementary Fig. 1), thus resulting in the DIC effect discussed in the following content.

Compared to conventional Au-PDMS and Au-SEBS devices, the fabricated Au-WPU device requires an extremely high peel strength of 1017.6 N/m to peel the Au layer from the WPU substrate (Fig. 1c), realizing strong anti-friction ability. To intuitively illustrate the anti-friction capability, the device is attached onto the human forearm skin for cyclic friction measurements. When rubbed 1000 times at a vertical pressure of roughly 65 kPa, the electrical resistance of the Au-WPU device only increases slightly from 8.7 Ω/□ to 35.7 Ω/□ (Fig. 1d). In comparison, the Au-PDMS and Au-SEBS devices are destroyed upon 5 and 70 frictions, respectively (Fig. 1e, f). These results demonstrate the superior anti-friction ability of the Au-WPU device, which is further verified by multiple friction experiments with the external loads applied by human fingerprinting (Supplementary Movie 1).

To further enhance the binding strength between Au and WPU layers, a rough surface configuration of WPU substrate with the roughness of 211.3 ± 3.4 nm is developed via a facile micropore-forming principle using the urea/water/N, N-dimethylformamide (DMF) solution ("Methods" section and Supplementary Figs. 4 and 5). The dissolved urea decomposes into ammonia and carbon dioxide bubbles at high temperature, which act as micropore-making agents[32]. The water-miscible DMF with a high boiling point and a slow evaporation rate reduces the volatilization rate of the solution, thus being able to modulate the microporous surface by controlling the nucleation and growth of bubbles[33,34]. The resultant Au-RPU device shows an extremely high peel strength of 1243.4 N/m, which is 100, 4 and 1.2 times higher than that of the Au-PDMS, Au-SEBS and Au-WPU devices, respectively (Fig. 1c). Here, the curve fluctuation in peel force test may be attributed to instrumental noise and binding non-uniformity[8], which can be alleviated by filtering and average value methods[8,35,36]. The peel strength of our Au-RPU device is significantly higher than that of other devices with inert metal functional layers (Supplementary

Table 2), confirming the advantages of proposed DIC strategy. In addition, the Au-RPU device features a microporous structure with the relatively large aperture diameter of ~101.6 ± 20.7 μm (larger than most of artificial skins with the diameter of a few microns or less), mimicking the human skin-like pores to enable the device breathable (Supplementary Fig. 6)[37].

Typical friction suffered in the metal layer of a device originates from the contact of the device and external objects, such as skin and fabrics, which can be quantified by repeatedly rubbing the surface of the device with a loaded force (Fig. 2a). Figure 2b and Supplementary Fig. 7 show the friction tests of typical Au-PDMS, Au-SEBS, Au-Ecoflex, Au-thermoplastic polyurethane (TPU), Au-polyimide (PI) devices and our Au-based devices by using a rough artificial aging skin under a 130 kPa vertical pressure to rub the device (see "Methods" section). Here, two reference sheet resistance (R) values of 500 Ω/□ and 100 MΩ/□ are chosen as the criteria of conductive and insulating states, respectively, as the device resistance is usually gradually increased during repeated rubbing. The number of frictions required to change the device from its conductive state to insulating state is only one time for the Au-PDMS device, and roughly five times for the Au-SEBS device, indicating that the Au layer is easily detached from the two elastic substrates. While the Au-WPU and Au-RPU devices can maintain highly conductive state after 28 ± 3 and 53 ± 5 frictions, and falls into the insulating state after 36 ± 2 and 61 ± 4 frictions, respectively (Fig. 2b), demonstrating the strong anti-friction ability. These results are further verified by scanning electron microscopy (SEM) images, in which the conductive Au films (bright area) are severely destroyed after one and five frictions for Au-PDMS and Au-SEBS devices, respectively; while the Au films are preserved with a few microcracks after 25 and 50 frictions for Au-WPU and Au-RPU devices, respectively (Fig. 2c). In addition to Au, Pt-RPU devices with the same fabricating process also exhibit excellent electrical tolerance to external frictions (Supplementary Table 3), indicating our strategy could be extended to other inert metallic materials.

We further verified that the Au-RPU device can stably keep highly conductive state against frictions under vertical pressure ranging from 65 kPa to 260 kPa (Supplementary Fig. 8). In addition, the influence of friction objects on the anti-friction ability of our device is characterized by using the artificial tender skin, artificial aging skin, woven fabric and abrasive paper that possess the increased surface roughness of 5.83 nm, 445.76 nm, 1.10 μm and 2.44 μm, respectively (Supplementary Table 4). The Au-RPU device can maintain substantially high electrical performance even after experiencing 1022 ± 76 frictions for smooth artificial tender skin, and 13 ± 2 frictions for highly rough abrasive paper (Fig. 2d). The quantitative comparison of peel strength and anti-friction results of these devices indicate our Au-RPU device has good electrical reliability as the artificial electronic skin. When applying a tensile test, the Au-RPU device can retain relatively good conductivity (resistivity: $1.09 \times 10^{-3}$ Ω m) even at a large stretchability of 400% (Fig. 2e), whereas, the other reported Au-based stretchable devices would lose their conductivity below 200% tensile strain (Supplementary Table 5). SEM studies reveal the interconnected conductive pathway with uniform Au micro-fragments for the Au-RPU device upon stretching (Supplementary Fig. 9). These micro-fragments can be recovered at 0% tensile strain without any noticeable slippage or exfoliation. In comparison, Au layer in the Au-PDMS device becomes disconnected fragments upon stretching, and slips clearly on the PDMS substrate (Supplementary Fig. 10). Although the Au-SEBS device exhibits good electrical performance at a large stretchability of 367% (Fig. 2e), localized slippage is still observed due to the weak interfacial binding (Supplementary Fig. 11). Together, these results indicate our Au-RPU device possesses stronger interfacial binding than Au-PDMS and Au-SEBS devices. The strong interfacial binding restricts the slippage of Au micro-fragments at the surface of RPU, thus achieving a high anti-friction ability. In addition, the Au-RPU device exhibits good

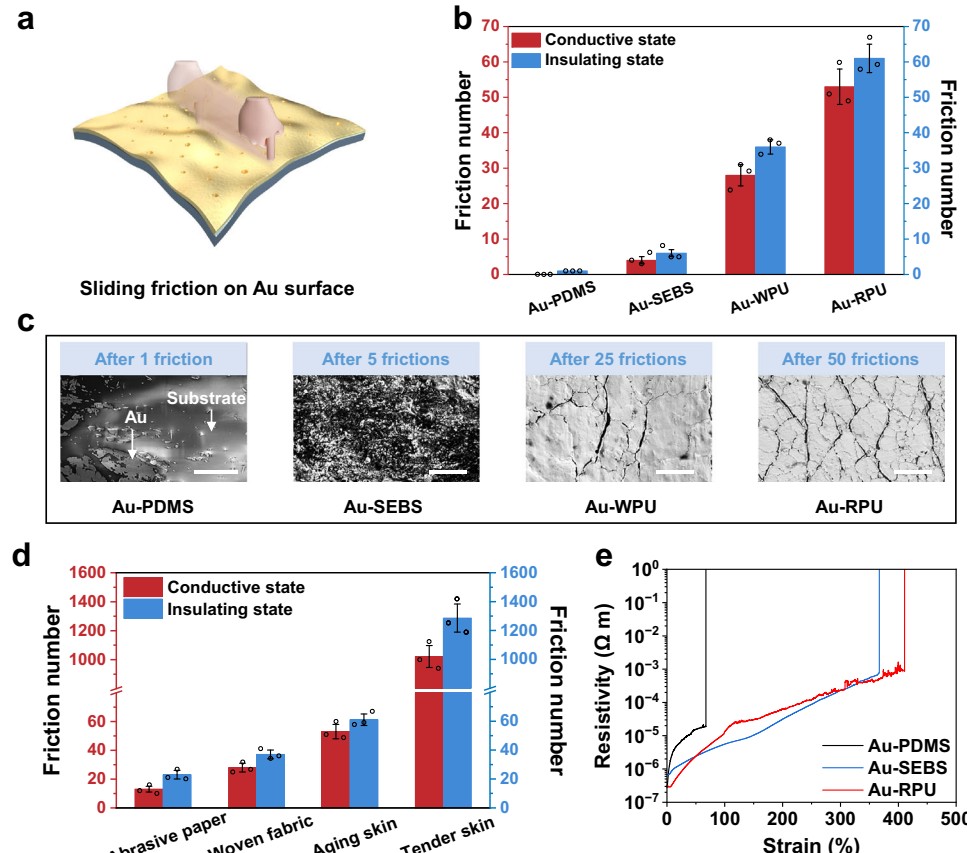

**Fig. 2 | Electrical anti-friction ability. a** Schematic illustration of Au-based stretchable devices upon cyclic frictions. **b** Anti-friction performance of Au-PDMS, Au-SEBS, Au-RPU and Au-RPU upon cyclic frictions using an artificial aging skin under 130 kPa pressure. **c** SEM images of Au-PDMS, Au-SEBS, Au-WPU, and Au-RPU after cyclic frictions. Bright area represents Au and dark area represents the substrate. Scale bar, from left to right: 200 μm, 10 μm, 3 μm and 3 μm. Each experiment was repeated three times independently with similar results. **d** Anti-friction performance of Au-RPU upon multiple frictions by four friction objects with the increased surface roughness. The vertical pressure is 130 kPa. **e** Stretchability comparison of the stretchable Au film on PDMS, SEBS and RPU substrates. Data in (**b**) and (**d**) are presented as mean values ± SD, $n = 3$ independent samples.

stability when applying five cyclic tensile deformation at different maximum tensile strains of 25%, 50% and 100% (Supplementary Fig. 12), and can retain 1000 stable tensile cycles at a maximum tensile strain of 100%, verifying its long-term durability upon large deformations (Supplementary Fig. 13).

## Mechanism analysis

To understand the interfacial binding nature of the Au/WPU interface, we used X-ray photoelectron spectroscopy (XPS) depth profiling to study the interface of the Au and WPU layer at each etched layer of Au-WPU sample (Supplementary Fig. 14 and Note 1). The XPS depth profiling shows that the Au 4$f$ signal intensity decreases whereas the C 1$s$ signal intensity increases obviously after the 9th etching of Au film, indicating the penetration of WPU into Au layer (Fig. 3a–c and Supplementary Table 6). Moreover, the remaining layer thickness of the coexistence of Au 4$f$ and O 1$s$ signals after the 9th etching is roughly 6.5 nm ("Methods" section and Supplementary Fig. 15), which is greater than the XPS electron escape depth of the O element at 1486.7 eV (~2.0 nm)[38,39]. These results indicate that O elements diffuse into the Au layer during the vacuum deposition.

To further explore the origin of O elements at the interface, the curve fitting of the O 1$s$ XPS spectra for surface uncleaned WPU and cleaned WPU were performed (see "Methods" section). Three fitting peaks, namely -OH, -COO- and C=O peak, are observed in the O 1$s$ spectrum of cleaned WPU, indicating the presence of oxygen-containing groups including hydroxyl and carbonyl groups on WPU

molecular chains (Fig. 3d, Supplementary Note 1). The relative atomic content of the -OH peak in the cleaned WPU is 39.46% (-OH groups on WPU molecular chains), but it increases to 50.50% in the uncleaned WPU (Fig. 3e and Supplementary Table 7). The results indicate that there should be also a certain amount of water molecules adsorbed on the uncleaned WPU surface, in addition to the oxygen-containing groups on WPU molecular chains. In comparison, no peak change is observed in O 1$s$ XPS spectra of the hydrophobic PDMS before and after surface cleaning (Supplementary Fig. 16), revealing that water molecules are prone to adsorption on WPU surface due to its intrinsic hydrophilicity[40]. Moreover, the fitting peak of -OH is dominated in the Au-WPU after 12th etching, which is consistent with the uncleaned WPU rather than the cleaned WPU (Fig. 3d–f, Supplementary Table 7). Together, these results demonstrate water molecules and oxygen-containing groups on WPU are capable of diffusing into the initially deposited Au layer during the deposition process.

Studies have demonstrated that clean Au surface is hydrophilic so that polar molecules like water molecules can wet Au surface[41,42]. The wetting effect induces Au grains to generate stable aggregates because a strong hydrogen bonding network would spontaneously construct among polar groups[43]. The hydrogen bonds were verified by in-situ Fourier transform infrared spectroscopy (FTIR; Fig. 3g, h; Supplementary Fig. 17). Two stretching vibration bands at 3385 and 3316 cm$^{-1}$ represent the ordered hydrogen bonding (H-bond) -OH and -NH- group, respectively. These bands move to a higher wavenumber upon heating, accompanied by the intensity reduction. These results

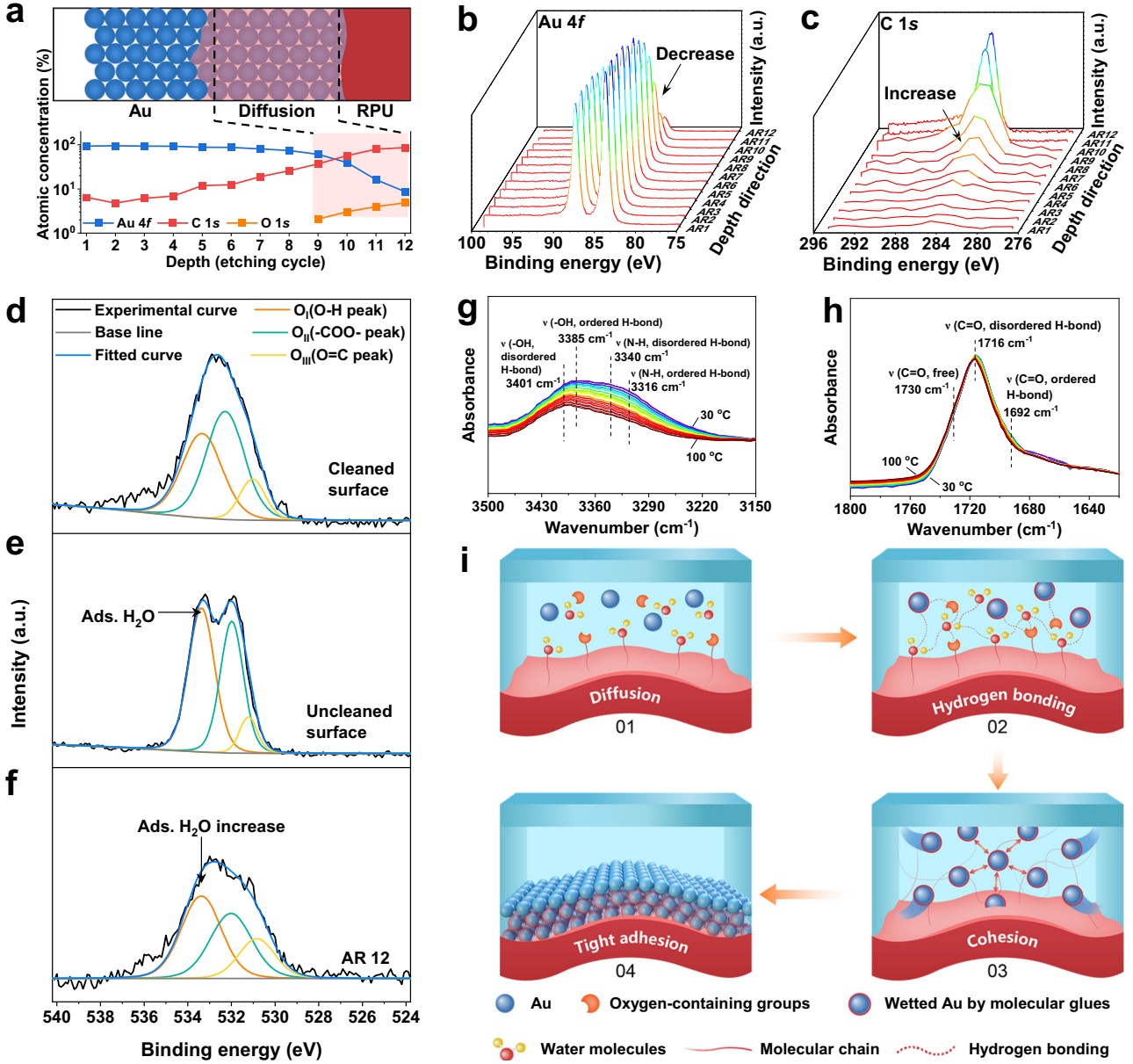

**Fig. 3 | Mechanism of the anti-friction Au-RPU device. a** Schematic (top) of the surface dense Au layer, the diffused Au layer and the WPU substrate at the interface and the corresponding atomic concentration from XPS (bottom) show Au grains at the Au-WPU interface are diffused into WPU. Shaded color with dashed black lines indicate the diffusion region. **b, c** Au 4f (**b**) and C 1s (**c**) XPS spectra of the Au layer on flat WPU substrate at different Ar⁺ ion etching levels. **d, e** O 1s XPS spectra of flat PU film **e** before and **d** after surface cleaning by Ar⁺ ion. **f** O 1s XPS spectra of the Au layer on flat WPU substrate after the 12th etching. **g, h** Temperature-dependent FTIR spectra of WPU upon heating from 30 °C (purple curve) to 100 °C (crimson curve): **g** 3500–3150 cm⁻¹; **h** 1800–1620 cm⁻¹. **i** Schematic illustration of the DIC mechanism for strong interfacial binding between Au layer and WPU substrate, including the diffusion of molecular glues, construction of hydrogen bonding, and cohesion deposition to achieve tight interfacial adhesion. Light blue and red color indicates the diffusion region and stretchable substrate, respectively. The abbreviated AR number in Fig. 3 is the number of times by Ar⁺ ion etching.

indicate that the ordered H-bond −OH and −NH− groups in WPU are transformed into disordered H-bond types (Fig. 3g). The C=O wide peak at 1716 cm⁻¹ is combined by the bands of free (1730 cm⁻¹), disordered H-bond (1716 cm⁻¹) and ordered H-bond C=O (1692 cm⁻¹). When the temperature is increased, the peak moves toward higher wavenumber with band intensity increasing for the free C=O and decreasing for the H-bond C=O groups, revealing the presence of hydrogen bonding in WPU. Notably, the peak intensity of disordered H-bond C=O is relatively stable with the slightly tendency of up-going first and then going down corresponding to the rearrangement of hydrogen bonds, which would be responsible to maintain the hydrogen bonding network[44–46].

Based on the above results, we propose a diffusion-induced cohesion effect to explain the strong binding of chemically inert Au on WPU substrate (Fig. 3i and Supplementary Fig. 1). In the evaporation process, water molecules and oxygen-containing groups on WPU surface act as molecular glues that bridge Au grains and WPU molecules. The molecular glues can diffuse into Au domains that reach the WPU surface and thus wet these Au grains at the initial stage of evaporation. A compact hydrogen bonding network is subsequently constructed among the molecular glues to create a strong cohesive force between molecules[47,48]. Such cohesive force can tightly pull adjacent wet Au grains together to be firmly anchored on the surface of WPU for anti-friction artificial skins.

Furthermore, we conducted finite element analysis (FEA) using the COMSOL software to explain the effect of the WPU surface morphology on the interfacial binding of Au layer (see "Methods" section). The surface geometry of the device is parameterized by using random numbers to generate rough surface for simulation (Supplementary Fig. 18a). When applying the same tangential force to the sample surface, the Au cell mesh on rough microporous WPU experiences less deformation stress than the flat WPU due to the large specific surface area of rough morphology (Supplementary Fig. 18). The simulation results reveal that rough topology structure can enhance the interface binding strength of the device for high anti-friction ability because of the increased effective contact area between the Au grains and the substrate.

## Anti-friction soft electrodes, circuits, and sensor array

In real HMIs, stretchable electronic devices are inevitably affected by the friction interference caused by the fabric, hair and skin, especially when people are doing intense exercises such as running and jumping. To demonstrate the feasibility of anti-friction Au-RPU electronics, we firstly fabricated Au-RPU-based stretchable electrodes for real-time human electrophysiological monitoring (Fig. 4a, see "Methods" section and Supplementary Movie 2 for measurement setup). Au-RPU stretchable electrodes were attached to a person's chest skin and cerebral cortex to monitor electrocardiograph (ECG) and electroencephalography (EEG) signals, respectively (Fig. 4b, c and Supplementary Fig. 19). High-fidelity ECG and EEG signals were continuously recorded before and after strong cyclic rubbing (Fig. 4d, Supplementary Movies 3 and 4), verifying the excellent anti-friction ability of our Au-RPU electrodes. In contrast, the commonly used Au-PDMS electrodes showed a significant deterioration in ECG and EEG signal recording after frictions due to the weak interfacial binding between Au and PDMS layer (Fig. 4d, Supplementary Movies 3 and 4).

Furthermore, we placed two Au-RPU electrodes on the forearm to monitor high-quality electromyography (EMG) recording while clenching the fists (Fig. 4e). The Au-RPU electrode showed a stable SNR of fist clenching before ($25.7 \pm 0.8$ dB), after 500 ($24.9 \pm 0.3$ dB) and 1000 ($25.0 \pm 0.9$ dB) frictions ("Methods" section and Fig. 4f). These results indicate our Au-RPU electrode shows excellent anti-friction ability for high-quality electrophysiological signal recoding, which is critical in tracking physical state of athletes during strenuous exercise.

We tested the interfacial impedance between skin and electrodes to evaluate the signal transduction ability of anti-friction electrodes (Fig. 4g). Compared with Ag/AgCl electrodes commonly used in clinic, the Au-RPU electrodes exhibit the lower interfacial contact impedance, which guarantees the signal integrity in electrophysiological measurement. More importantly, the Au-RPU electrodes still could maintain a low skin interfacial impedance even after rubbing on the forearm skin for 1000 times with roughly 65 kPa. The electrode impedance after frictions is slightly higher than that of the original electrode in low-frequency range of $10^0$-$10^2$ Hz, which may be related to the resistance change of the device after the friction[10].

We also utilized the patterned Au-RPU film to fabricate an artificial skin circuit consisting of Au-RPU-based interconnect wires and four LEDs (Fig. 4h). To demonstrate the electrical anti-friction performance, the stretchable circuit was rubbed by using human skin and fabrics that are conventional friction sources in daily use. The four LEDs in the circuit were lit simultaneously and maintained almost constant brightness upon multiple frictions (Supplementary Movie 5). Furthermore, the circuit worked well upon stretching, folding, and kneading deformations (Fig. 4i–k, Supplementary Movie 6). These results demonstrate that Au-RPU interconnect wires possess strong anti-friction ability at the circuit level for the applications of anti-friction artificial skins.

The next generation of bionic soft robots are expected to enable active interactions with surrounding environment through large-scale force acquisition and feedback (Fig. 5a). However, the accuracy and reliability of force acquisition and feedback are often limited by the friction interference of sensing array[49,50]. To solve this issue, we fabricated an anti-friction pressure sensor array (APSA) based on the Au-RPU electrodes, which could be attached to sharp 3D surfaces and reliably perceive the pressure signals up to 22.2 MPa when subjected to concentrated deformation stresses (Fig. 5b). The APSA consists of the force-sensitive interlayer addressed by an orthogonal network of conductive Au electrodes deposited on the RPU substrates ("Methods" section, Fig. 5b and Supplementary Fig. 20). There are 36 sensors in the array (3 rows × 12 columns), and each point of overlap between the orthogonal Au electrodes is a single sensing unit that is sensitive to the normal pressure. As for a single sensing unit, the absolute value of the relative change in electrical resistance, $|R-R_0|/R_0$, increased by near 80% when applying 22.2 MPa pressure (Fig. 5c). The force response within 0.4 MPa to 22.2 MPa exhibited consistent behavior across multiple sensing units (Supplementary Fig. 21), which is superior to most of existing inert metallic electrodes-based pressure sensors (Supplementary Table 8). In addition, the APSA sensor unit shows good stability at the temperature up to 60 °C, 100% relative humidity, 20% tensile strain for electrodes or 90° bending condition (Supplementary Fig. 22). The sensing value of each individual sensor in the array could be correctly extracted by using a ground-based electrical isolation scheme (Supplementary Fig. 23).

To further illustrate the superiority of anti-friction artificial skin, our APSA was attached to a person's palm to record the magnitude and distribution of pressures when grasping objects with different surface sharpness (Fig. 5d–i). When grasping a smooth apple (weight: roughly 250 g), our APSA can obtain a uniformly varied pressure mapping due to the conformable contact between the apple and the APSA (Fig. 5d–f and Supplementary Movie 7). However, when the surface of object becomes sharp, i.e., grasping a melon (weight: roughly 250 g) with the horn-like protrusion skin, the sensor array could not conformably contact the object, resulting in concentrated pressures to the APSA (Fig. 5g, h and Supplementary Movies 8 and 9). The recording pixels on the sharp concentrated deformation domains are the largest in the pressure mapping, and gradually decrease toward the adjacent regions (Fig. 5i). In contrast, a fabricated pressure sensor array based on the Au-PDMS electrodes could not record the pressure mapping when grasping the horned melon and the smooth apple (Supplementary Fig. 24). This is because most of the patterned Au layers on the PDMS substrate would be scratched off when the array is subjected to a small pressing force and deformation (Supplementary Movie 10) due to the poor interfacial anti-friction ability. These results demonstrate our APSA have the excellent large-scale and concentrated force acquisition ability for anti-friction HMI applications, exhibiting superiority to most inert metallic electrodes-based pressure sensor arrays (Supplementary Table 8). To further verify the cyclic stability under concentrated force, our APSA was conducted on sharp deformation tests at a maximum pressure of 2.2 MPa pressure. There is no obvious change in the recording signal amplitude after 200 cycles of sharp deformation (Fig. 5j). Meanwhile, the delamination of the Au electrode layer from the RPU substrate was not observed from SEM results after 200 cycles, further verifying the excellent interfacial anti-friction capability of our APSA (Fig. 5k). In addition, our APSA could be attached to the fist to accurately and reliably record the pressure mapping during punching exercise (Supplementary Figs. 25, 26 and Movie 11). These results demonstrate our anti-friction APSA enables reliable and accurate recording of concentrated large forces for motion detection, rehabilitation medicine and soft robotic applications.

## Discussion

In summary, we report a simple and effective DIC approach without complex chemical modification to create anti-friction artificial skins for HMIs. The approach involves the diffusion of molecular glues into Au layer to induce hydrogen bonding cohesion effect, leading to the

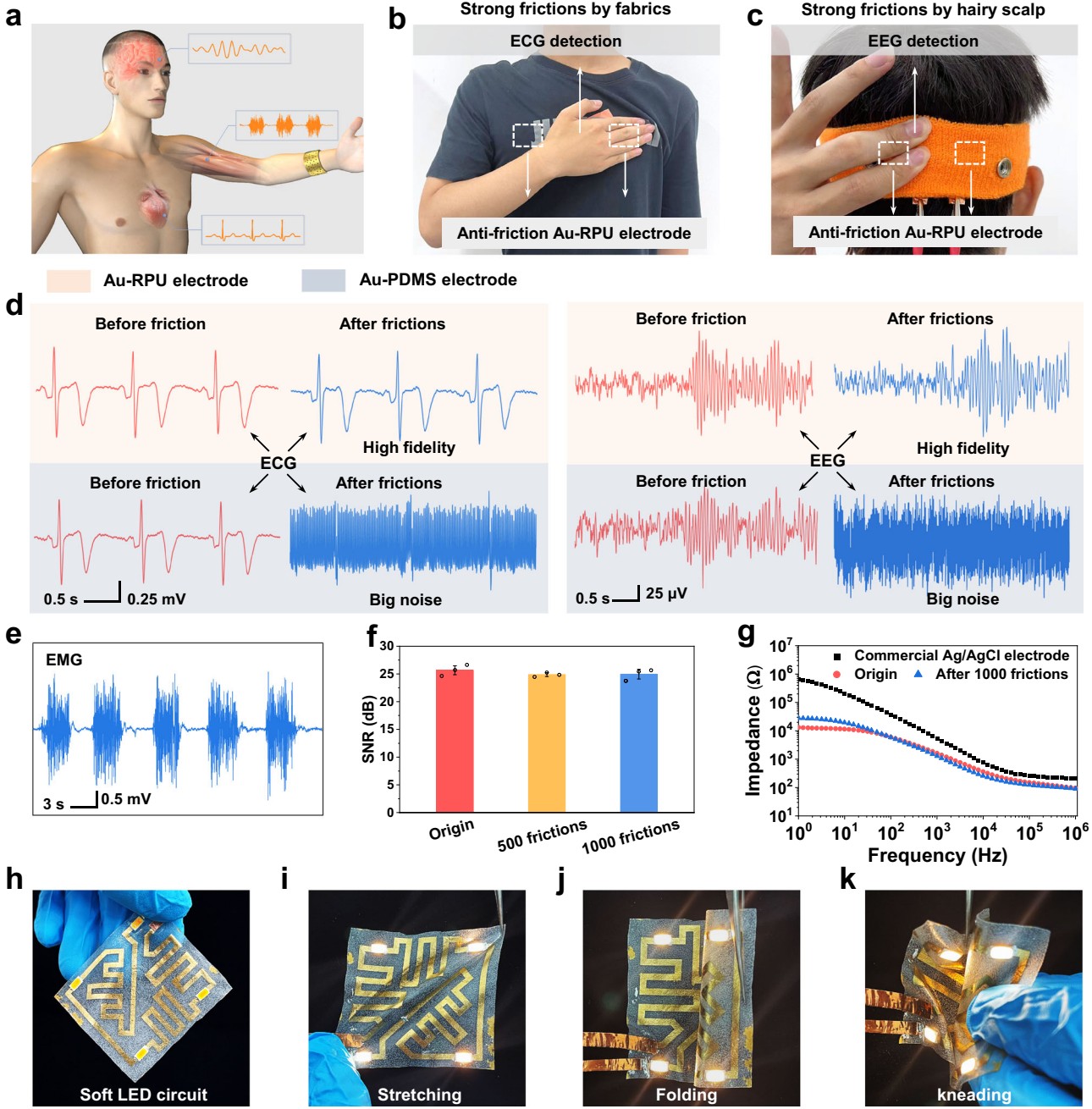

**Fig. 4 | Anti-friction stretchable electrodes and circuits. a** Schematic illustration of soft electrodes for reliable human biosignal recording. **b** Optical image of ECG recording for a male volunteer upon strong cyclic rubbing by fabrics. The dashed white area indicates the soft electrodes position. **c** Optical image of EEG recording for a male volunteer upon strong cyclic rubbing by hairy scalp. **d** Comparison of recording ECG and EEG signals when using Au-RPU and Au-PDMS electrodes before and after cyclic frictions. The faint red area indicates the test data from Au-RPU electrodes, while the light gray area indicates the test data from Au-PDMS electrodes. The data before and after fricitons are labeled by red and blue color. **e** Reliable and high-quality EMG signals recorded by Au-RPU electrodes even after 1000 frictions on the human forearm. **f** Comparison of the SNR values recorded with Au-RPU electrodes before and after 500 and 1000 cyclic frictions. **g** Comparison of electrical impedance spectroscopy measurements between the human skin and commercial Ag/AgCl electrode, as-deposited Au-RPU electrode, and Au-RPU electrodes after 1000 frictions. **h–k** Optical image of the stretchable LED circuit based on patterned Au-based interconnecting wires with the line width of 2 mm on RPU substrate (**h**) for reliable operation when stretching (**i**), folding (**j**) and kneading (**k**). Data in (**f**) are presented as mean values ± SD, $n = 3$ independent samples.

binding of chemically inert Au grains to an elastic waterborne polyurethane with a strong interfacial adhesion of 1017.6 N/m. This interfacial adhesion is further improved to 1243.4 N/m by constructing a nanoscale rough surface configuration. Consequently, the Au-RPU device can maintain good conductivity even after $1022 \pm 76$ frictions at 130 kPa pressure. Moreover, the device exhibits a low resistivity ($1.09 \times 10^{-3}\ \Omega\,m$) under 400% tensile strain, and almost recovers to its original state completely without any noticeable slippage or exfoliation of the Au flakes. To evaluate the practical utility of our anti-friction artificial skins, we fabricated soft electrodes that can record high-fidelity electrophysiological signals even after strong cyclic rubbing by fabrics, hairy scalp and skins. We also integrated the Au-RPU interconnect wires into a pressure sensor array to reliably collect concentrated large pressures up to 22.2 MPa during grasping activities. Our findings provide a chemical modification-free solution to the challenge of weak interfacial binding for inert metal-based stretchable

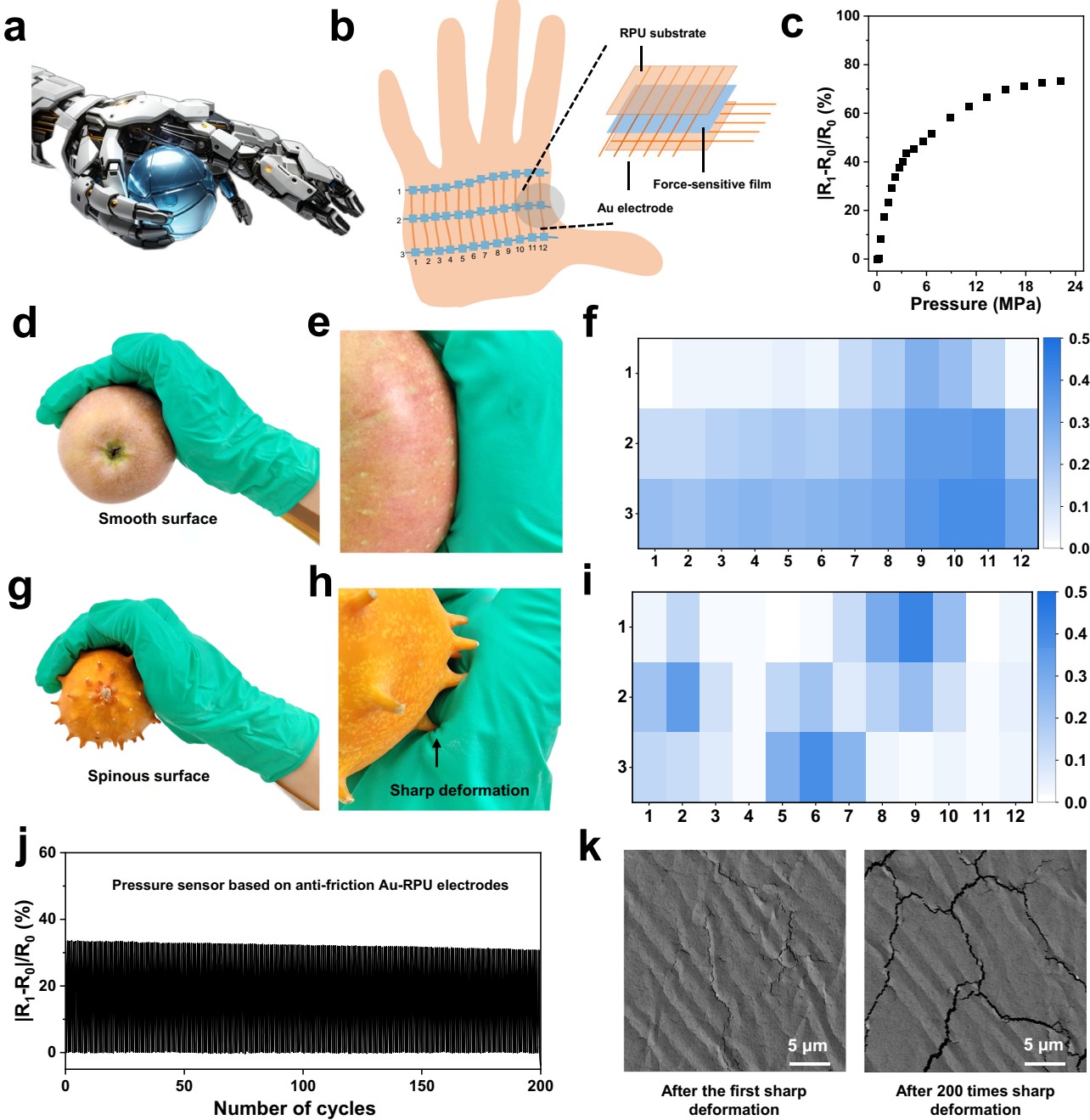

**Fig. 5 | Anti-friction APSA. a** Schematic illustration of a bionic robot palm integrated with tactile perception to grasp an object. **b** Schematic of structure and location of the APSA with 36 sensors (3 rows × 12 columns) and 15 electrodes. **c** Response of the sensing unit of APSA before and after 1000 frictions. **d** Optical image of the APSA attached on the palm of a female volunteer to grasp an apple with smooth surface. **e** Zoom-in image showing the conformable contact between grasper and the apple. **f** Pressure force maps of the APSA with a color scale indicating the pressure when grasping the apple. **g** Optical image of the APSA attached on the palm of a female volunteer to grasp a horned melon with spinous surface. **h** Zoom-in image showing unconformable contact between grasper and the horned melon with sharp deformation domains. **i** Pressure force maps of the APSA with a color scale indicating the pressure when grasping the horned melon. **j** Stability of the APSA during cyclic sharp deformation tests under 2.2 MPa pressure. **k** SEM images of Au-RPU electrodes used for the APSA after the first (left) and 200 times (right) sharp deformation test under 2.2 MPa pressure. Scale bar: 5 μm. Each experiment was repeated three times independently with similar results.

electronics, enabling the potential applications of anti-friction artificial skins in robust wearable and HMI systems.

## Methods

### Ethics approval

All human subject studies were approved by the Ethics Committee of Zhongshan Hospital, Fudan University. ID: Y2021073. Informed consent was obtained from all participants involved in this entire study.

### Materials

Waterborne polyurethane emulsion (Impranil®, mass content: 50 wt%) was purchased from Covestro; SEBS (Tuftec H1221) was purchased from Asahi Kasei; Ecoflex elatomer 00-30 (Smooth-On, Inc) was prepared by mixing Ecoflex component A and B, 1:1 by volume; PDMS GYLGARD 184 was purchased from Dow Corning; TPU film, PI film, urea, tetrahydrofuran (THF, ≥99.9%), and N, N-Dimethylformamide (DMF, ≥99%) were purchased from Sigma-

Aldrich. All these materials were used directly without any further treatment.

**Fabrication of RPU substrate.** To achieve the RPU substrate with a rough surface configuration, a certain amount of urea was added into 5 mL deionized water, followed by adding the WPU emulsion and stirred for 30 min to obtain a dispersed solution. Next, DMF and deionized water were added into the solution and stirred for 1 h to obtain the final WPU-urea-DMF mixture solution. Then, 5 g mixture solution was poured into a Teflon petri dish (7 cm in diameter) and placed onto a heating table in the fume cupboard at 100 °C for 30 min to obtain a semi-dry microporous film. Next, the film with Teflon petri dish was immersed in deionized water for 12 h, and the deionized water was changed every 3 h to remove the residual urea. The soaked film was dried at 40 °C for 24 h to obtain the final RPU elastic substrate. Samples with different urea contents were prepared using the same procedures. To prepare the WPU elastomer with an aperture diameter of roughly 100 μm, the optimized mass ratio of 24.0:1.2:24.8:50 for WPU-urea-water-DMF was used. For PDMS control sample, PDMS solution was prepared by mixing the base and curing agent with a mass ratio of 10:1. After removing bubbles, a certain amount of mixture solution was poured into glass petri dish and dried at 60 °C for 10 h to obtain the PDMS elastic substrate. For SEBS control sample, 10 g SEBS particles were added to 45 g THF solvent and stirred for 24 h to obtain the homogeneous solution. A certain amount of solution was then poured in glass petri dish and dried in a fume hood at room temperature for 24 h to obtain the SEBS elastic substrate.

**Fabrication of anti-friction soft electrodes and LEDs circuit.** For anti-friction Au-RPU electrodes, a 50-nm-thick Au was deposited on as-prepared RPU substrate through e-beam evaporation (DE400, DE Technology). Au-PDMS and Au-SEBS electrodes were prepared with the same deposition condition. Unless otherwise noted, 50-nm-thick Au was used in the experiments. ECG, EEG and EMG tests were performed using a commercial readout circuit (Backyard Brain Spikerbox) connected to a notebook. Both male and female volunteers participated in those electrophysiological signal test. EMG measurements were taken in the musculocutaneous area of the right forearm. Conductive copper tape was used to connect the electrodes to the readout circuit. For anti-friction LEDs circuit, a patterned Au layer was deposited on the RPU substrate as interconnect wires via the shadow mask technology during evaporation. The line width of Au-based interconnect wires was 2 mm. Four patch LEDs (3 mm × 5.6 mm) were then soldered onto the circuit using liquid metal EGaIn (Sigma 495425).

**Fabrication and read out of anti-friction APSA.** The sensing unit was consisted of the commercial force-sensitive film layer (3 M Velostat electrically conductive polymer film purchased from Adafruit Industries) and the carbon conductive adhesive layer. The conductive carbon adhesive layer was sandwiched between the force-sensitive films in order to increase the force-sensing range of the sensor. The patterned Au-RPU interconnect wires with 1.5 mm line width were used as top and bottom electrodes. A $3 \times 12$ pressure sensor array was fabricated by stacking the bottom interconnect wires, sensitive units, and top interconnect wires. No additional adhesives were required for encapsulation due to the binding effect between electrode layers. A custom flexible printed interconnect wires were used to interface the sensor array with a custom readout circuit. The typical sensor force response was measured by applying controlled normal pressures using a mechanical tester (MTS Systems C42) and simultaneously monitoring the signals using a DAQ6510 Data Acquisition Multimeter System (Keithley Instruments). As for the sensor array based on Au-PDMS electrodes, the device fabrication was exactly the same as above except that the RPU substrates were replaced by PDMS substrates.

The custom readout circuit was design to alleviate the crosstalk issue of the passive pressure sensor array based on an electrical-grounding-based readout mechanism. In this approach, the row of the selected sensor was grounded while all other rows were maintained at the reference voltage $V_{ref}$ (2.5 V in our design). The current flowed through all unselected sensors was zero since the voltage difference across them was zero. A 4:1 analog switch was used as an analog demultiplexer to raster through the row and read out individual resistances one by one. The 16 single-pole double throw switches were applied to ground the next row while returning the selected row to $V_{ref}$. The single output of the analog switch was converted to a digital signal (10-bit resolution; 0−1023 corresponding to 0−5 V) and the array data is transmitted to the computer in real-time based on MATLAB software.

**XPS depth profile analysis.** XPS depth profiles was performed on 20-nm-thickness Au/flat WPU sample by using a K-alpha X-ray photo-electron spectrometer (Thermo Fisher Scientific, Waltham, MA), having an Al Kα X-ray source (1486.7 eV) and a 3.8 kV Ar$^+$ ion gun. The raster size was 3 mm × 3 mm, the etch step time was 15 s, and the energy step size was 0.05 eV. Each spectrum was acquired as an average of five scans. As for the surface cleaned WPU and PDMS substrates, Ar$^+$ ion etching was used to clean the adsorbed molecules on WPU and PDMS surfaces. The surfaces of uncleaned WPU and PDMS were not cleaned by Ar$^+$ ion to reflect the actual chemical state of the substrate surface during evaporation. XPS chemical trace analysis was performed using CasaXPS software.

**Temperature-dependent FTIR spectroscopy.** WPU film was put on an in-situ heating platform. Then, the film was heated from 30 °C to 100 °C at 2 °C min$^{-1}$, and the temperature-dependent FTIR-ATR spectra within the region 3600−600 cm$^{-1}$ were collected at the same time. A Nicolet iS50 Fourier transform spectrometer equipped with a DTGS detector was used for the FTIR measurements. All the FTIR spectra were gathered by 20 scans with a resolution of 4 cm$^{-1}$ to obtain a good signal-to-noise ratio.

**Finite elemental analysis.** The stress and deformation analysis of Au layer on flat WPU and rough Au-WPU were conducted by using CMO-SOL software. The model size of 10 μm × 10 μm × 2 μm (Au layer: 0.1 μm, substrate layer: 1.9 μm) were used for both samples. For the rough surface model, parameterized surfaces were constructed using random numbers to generate rough surfaces. For the flat surface model, the interface between Au layer and substrate was not roughened. As for interfacial adhesion simulation, the substrate was constrained, and the X-direction stress was applied to the surface of Au layer to produce a relative displacement between the gold layer and the substrate. Next, the stress distribution of the interface layer was investigated. The values of the material parameters involved in the analysis were as follows: Young's modulus 1.3 MPa, Poisson's ratio 0.38, density 1.1 g·cm$^{-3}$.

**Mechanical characterization.** The sample was cut into 15 mm × 10 mm for the test. Test samples were fabricated by taping 3 M 5112 double-sided tape on Au layer and holding for 1 h. T-peel test was measured using a tensile test machine (MTS Systems C42, 250 N load-cell) with the tensile speed of 5 mm/min. Peel strength was calculated by dividing the plateau force by the width of the sample.

**Electrical performance testing.** To measure electrical anti-friction ability, each sample with a 20 mm in length and 10 mm in width was prepared and adhered to the platform using a layer of double-sided tape to ensure stably test. The vertical loads were controlled using a force gauge with a universal test machine (SH-50N, Nscing Es Mechanical Testing Co., Ltd). Different friction objects were attached

to a circular tray (10 mm diameter) by using the double-sided tape (3 M 5112) to apply frictions. The artificial aging skin was prepared by mixing PDMS base and curing agent with mass ratio of 3:1. When PDMS was in semi-cured state, the finger fingerprint was imprinted into the PDMS to ensure the high surface roughness. The artificial tender skin was prepared by mixing PDMS base and curing agent with mass ratio of 15:1. When PDMS was in semi-cured state, smooth forearm skin was imprinted into the PDMS to ensure the low surface roughness. As for friction tests, 65 kPa, 130 kPa, and 260 kPa vertical pressures were applied to each sample and moved 10 mm along the sample width. The electrical performance of the device was real-time recorded using a DAQ6510 Data Acquisition Multimeter System (Keithley Instruments). The cyclic tensile/release tests were measured using the DAQ6510 equipment and a mechanical tester (MTS Systems C42). The repeated sharp deformation tests were measured using the DAQ6510 equipment and mechanical test machine (Mark-10). The pressure sensor was heated to 40, 50 and 60 °C by a hot plate and held for 5 min. Ultrasonic humidifier was used in a chamber to create a high-humidity condition (relative humidity: 100%) for the pressure sensor. As for mechanical deformations, the sensor was tested under 20% tensile strain or 90° bending condition. The cyclic friction was counted until the device changed from a high conductance state to an insulation state.

**Skin interfacial contact impedance test and SNR calculation.** The interfacial contact impedance was measured by attaching pairs of electrode with a square shape (side length: 1.5 cm), and a center-to-center distance of 5 cm on forearm skin. The tests were conducted using electrochemical workstation (GAMRY Interface 1010) from $10^6$ to $10^0$ Hz with the voltage of 100 mV. The EMG, ECG, and EEG signals were recorded by a commercial printed circuit board (Backyard Brain). SNR of EMG signals were calculated as following equation:

$$SNR = 10\log_{10}\left(\frac{P_{signal} - P_{noise}}{P_{noise}}\right) \qquad (1)$$

$P_{signal}$ and $P_{noise}$ denote the mean power of the signal and noise, respectively. They were estimated from the obtained signals according to the reported algorithm[51].

**Reporting summary**

Further information on research design is available in the Nature Portfolio Reporting Summary linked to this article.

## Data availability

The source data generated in this study are provided in the Supplementary Information/Source Data file. Source data are provided with this paper.

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

## Acknowledgements

This work was supported by the National Key Research and Development Program of China: 2021YFB3601200 (M. W.), National Natural Science Foundation of China: 62204052 (J. C.), 62104042 (M. W.), 62204057 (E. S.) and 62074164 (G. X.), Shanghai Pujiang Programme: 23PJD003 (J. C.), the STI 2030-Major Project No. 2022ZD0209900 (E. S.), and Lingang Laboratory: LGQS-202202-02 (E. S.).

## Author contributions

J.C. and M.W. designed the study. J.C. designed, fabricated, and characterized the devices. S.X.L. and J.C. conducted the FEA simulation. F.Z.Y., J.Q., J.C., and Y.L. fabricated and characterized the APSA signal readout circuit. J.C., Y.L., Q.X., Y.C., J.W.C., and E.M.S. recorded the optical images and videos. J.C. and M.W. wrote the paper. H.F.C., G.Z.X., E.M.S., Q.L., and M.L. revised the original draft. M.W., Q.L., and M.L. supervised the whole work.

## Competing interests

The authors declare no competing interests.
