## [Peer Review File · Nature Communications]

Reviewers' comments:

Reviewer #1 (Remarks to the Author):

This manuscript reports an interfacial diffusion-induced cohesion strategy based on hydrophilic polyurethane to wet gold (Au) grains and nanoscale rough configuration of the polyurethane (RPU) to result in a high interfacial binding strength of over 1000 N/m. With much higher binding strength than that of conventional polydimethylsiloxane and styrene-ethylene-butylene-styrene, the stretchable Au-RPU device can remain good electrical conductivity after one thousand frictions with a 10 N vertical load, and reliably record high-fidelity electrophysiological signals. The demonstration also includes an anti-friction pressure sensor array with Au-RPU interconnect wires. However, the unique advantages of the strategy developed in this work are not explicitly leveraged to address niche applications that are challenging with alternative approaches. The authors may also want to address the following comments before consideration for publication.

1. There have been extensive studies on interface designs based on many other materials (beyond PDMS and SEBS) for wearable sensors and devices. How does the result reported in this work compare with the others in the literature?
2. Anti-friction artificial skins also do not need to be limited to the use of Au, so the comparison and discussion with the other conducting materials should be made.
3. How does the anti-friction performance of the Au-RPU device depend on the pattern and geometric parameters of the rough configuration?
4. How does the cyclic friction in Fig. 4b correlate to the use of the sensors in practical applications? Many previously reported sensors have already shown the robust performance of electrophysiological signal detections during daily use or even in extreme use cases. Therefore, it is important to provide the demonstrations/applications that can be challenging with the other reported sensors.
5. The validation and application of the EEG and EMG signals can be helpful to include.
6. The Au-RPU electrode exhibits much lower contact impedance than that of the commercial gel electrodes, but gives a signal-to-noise ratio lower than many other published results (that do not even show superior contact impedance). Could the authors comment and explain?
7. The comparison of the fibrous pressure sensors in Table S1 is quite limited. Please benchmark the complete performance parameters (e.g., sensitivity, linearity, the limit and range of detection, and long-term stability, among others) of the sensor against those in the literature to clearly demonstrate the novelty. The comparison can also go beyond the fibrous sensors to include the other pressure sensors.
8. The use of the anti-friction property is not clearly outlined in the pressure/force sensing (Fig. 8), as many previous reports have shown reliable sensing "after 1000 pressing deformations".
9. How does the sensor performance change with variations in environmental conditions (e.g., temperature and highly humid or water moisture environments) or with mechanical deformations (e.g., stretching, bending, or twisting)?

10. The introduction section may also be updated to include the state-of-the-art literature on fabrication methods to prepare deformable sensors or functional circuits on 3D freeform surfaces, multimodal sensors with decoupled sensing mechanisms, and standalone stretchable sensing platforms to provide comprehensive background and future opportunities.

Reviewer #2 (Remarks to the Author):

The method involves using hydrophilic PU to coat the gold grains, creating strong hydrogen bonding and resulting in a high interfacial binding strength. Through an examination of the friction resistance of the stretchable electrodes, the authors have demonstrated the feasibility of detecting human bio-signals and achieving circuit interconnects. However, it is worth noting that while the stretchable electrodes exhibit increased mechanical resistance against external rubbing forces, their mechanical stretchability remains relatively poor. This limitation could potentially restrict its applications in soft electronics. Additionally, the manuscript lacks dedicated in-depth analysis and demonstrations, which are necessary for publication in a prestigious journal like Nature Communications. Therefore, I do not support the publication of this paper in its current form. Below are detailed comments addressing specific aspects of the manuscript.

1. The primary focus of this research is to enhance the mechanical integrity of gold-based stretchable interconnects, particularly developing the electrodes more durable to external stimuli (i.e., tangentially rubbing under a given force). Despite achieving remarkable improvements in the adhesion properties of the Au thin film via increasing the interfacial binding strength, the authors have not been able to maintain the electrical conductivity of the interconnects when subjected to mechanical strain, which is a crucial requirement for their development in more practical applications. In specific, the authors claim that the Au-RPU device exhibits a low resistivity ($1.09 \times 10^{-3} \Omega \text{ m}$) even at a large stretchability of 400%. Not only the substantial increase of (approximately fourfold) the resistivity upon 400% mechanical stretching is expected to hinder the applicability of the stretchable interconnects but also a relatively low friction number (max of 1022) achieved in this study is considered insufficient performance compared to previously reported research findings. Furthermore, the authors did not conduct comprehensive stretchability tests.

2. The authors primarily focused on comparing the performance and mechanical durability of the stretchable interconnects with PDMS and SBS-based stretchable electrodes. However, it is evident that the Au-RPU interconnects would undoubtedly exhibit superior electrical and mechanical properties due to their enhanced interfacial binding strength. To establish the applicability of stretchable interconnects in a wide range of soft electronics, it would be more reasonable to compare them to previously successful demonstrations of gold-based stretchable interconnects, such as those published in Nature (500, 59-63, 2013) and Nature Nanotechnology (12, 907-913, 2017). This approach would provide a more solid basis for claiming the excellence of the anti-friction gold-based stretchable electrodes.

3. It is considered oversimplified to merely demonstrate human bio-signals and the electrical response to a certain level of impact as a measure of the development of stretchable electrodes. Specifically, the manuscript lacks the presentation of an electrical circuit setup that enables real-time monitoring of signal changes in response to mechanical impact. It is crucial that this construction of the electrical

circuit be shown in a video or supplementary material to provide a more comprehensive understanding of the research.

4. It is necessary to express all electrical performance measurements in terms of sheet resistance (ohm/sq.). Additionally, the normal loads should be presented as pressure values based on the Hertzian contact model. This approach is crucial for facilitating clear comparisons and ensuring a comprehensive understanding of the level of pressure applied in the experiments.

Reviewer #3 (Remarks to the Author):

Recommendation: Publish in Nature Communications after major revisions noted

Comments:

The manuscript entitled "Anti-friction gold-based stretchable electronics enabled by interfacial diffusion-induced cohesion" submitted by Ming Liu and coworkers presented a chemical modification-free diffusion-induced cohesion (DIC) strategy to achieve strong interfacial binding between inert Au and waterborne polyurethane (WPU) for anti-friction electronics. While the material design presented in this work is intriguing, there are several unsupported claims regarding the anti-friction ability of Au-RPU throughout the figures and text. 1) Given the potential influence of temperature and humidity on the functional groups and molecular interactions within the composite materials, the authors should provide a range of conditions that ensure consistent performance. 2) In addition, the mechanical mismatch between solid inert metals and elastomers significantly affects interfacial defects; however, the modulus of the materials is not addressed in the manuscript. Because of these two reasons, the impact of this work may be limited. Overall, I recommend the addition of more experiments and appropriate discussion for the aforementioned issues. Here are some detailed comments:

1) The authors mentioned the "cohesion effect of adjacent wetted Au grains in the interfacial layer between Au and PU." While I understand the author's intention behind this concept, the definitions of molecular interactions and hydrogen bonds occurring between Au and PU are not clear. Additionally, the exact molecular structures are not provided, making it challenging to comprehend the notion of "molecular glues." If the authors could provide a more explicit representation or description schematic of the molecular structure involved in the Au-PU interface, it will help readers to better understand this work

2) In Figure 1c, the authors measured the peel strength as a function of distance for Au-WPU and Au-RPU, and these values fluctuate significantly. Given the envisaged use of these sensors in scenarios involving bending or movement, such fluctuations cause undesirable noise, thereby reducing the signal-to-noise ratio. Consequently, it is imperative for the authors to provide a comprehensive discussion to mitigate these concerns and offer suitable solutions.

3) In Figure 1, The authors provided XPS results for polyurethane(PU). While the authors provided the XPS result of PDMS, there are no C 1s peaks or Au 4f peaks, or any SEBS peaks that can be compared with those of PU. In addition, it may be inaccurate to identify the molecular interaction between Au, Pu, and water solely through the increased intensity observed in the XPS results. To strengthen the author's claim, it would be valuable to include additional evidence and analysis. In my opinion, exploring

alternative approaches or techniques that can offer more direct information of hydrogen bonds, would be beneficial. The provided reference could provide insight into analyzing peak shift in XPS in terms of hydrogen bonding.

Ref: Lei, Y., Tang, Z., Zhu, L., Guo, B., & Jia, D. (2011). Functional thiol ionic liquids as novel interfacial modifiers in SBR/HNTs composites. *Polymer*, 52(5), 1337-1344.

4) In Figure 4c, the impedance value of the friction sample is higher than the original in the low-frequency region, but lower in the high-frequency region. It is required to define the impedance relaxation behavior according to frequency and provide an explanation for the observed differences in impedance at low (< 10 Hz) and high (> 105 Hz) frequency regions.

5) In Figure 3a, there is a typographical error in the y-axis label. The word "Aomic" should be corrected to "Atomic."

Detailed responses to reviewers' comments

Reviewer #1:

Comments: *This manuscript reports an interfacial diffusion-induced cohesion strategy based on hydrophilic polyurethane to wet gold (Au) grains and nanoscale rough configuration of the polyurethane (RPU) to result in a high interfacial binding strength of over 1000 N/m. With much higher binding strength than that of conventional polydimethylsiloxane and styrene-ethylene-butylene-styrene, the stretchable Au-RPU device can remain good electrical conductivity after one thousand frictions with a 10 N vertical load, and reliably record high-fidelity electrophysiological signals. The demonstration also includes an anti-friction pressure sensor array with Au-RPU interconnect wires. However, the unique advantages of the strategy developed in this work are not explicitly leveraged to address niche applications that are challenging with alternative approaches. The authors may also want to address the following comments before consideration for publication.*

Response: Thank you for your valuable comments to improve the quality of this work significantly. In the revised manuscript, we have addressed the reviewer's concerns with additional data as shown below.

1. We have added three new demos to leverage the unique advantages of our anti-friction Au-RPU devices prepared by DIC strategy over other strategies. In the first two demos, our anti-friction Au-RPU electrodes and conventional Au-PDMS electrodes were attached to chest skin and hairy scalp, respectively, and the captured electrophysiological signals before and after dynamic rubbing were used for comparison (**Figure 4 and Supplementary Movie 3 and 4** in the revised manuscript). In the third demo, a newly assembled pressure sensor array based on the anti-friction Au-RPU electrodes was attached to 3D surfaces with different sharpness, which could reliably perceive force signals up to 22.2 MPa even when subjected to concentrated deformation stresses (**Figure 5 and Supplementary Movie 7-9** in the revised manuscript). We believe these demonstrations will be useful to elaborate the unique advantages of our strategy.

2. We have also added several comparisons and descriptions on the interfacial binding strength of metal-based electronic devices (**Supplementary Table 2**), electrical properties and anti-friction abilities of Au-based stretchable devices (**Supplementary Table 3 and 5**), and comprehensive performances of metal electrode-based pressure sensors (**Supplementary Table 8**), to verify the superior performance of our Au-RPU-based devices.

In addition, we have added many other experimental data to carefully address the reviewer's technical comments. Figures and descriptions are added in the revised manuscript.

Q1: *There have been extensive studies on interface designs based on many other materials (beyond PDMS and SEBS) for wearable sensors and devices. How does the result reported in this work compare with the others in the literature?*

Response: Thank you for your comment. We supplement the comparison of the

interfacial binding strength of our device with those reported in the literature (based on tape-peel tests), as shown in **Supplementary Table 2**. Generally, there are two typical methods to test the interfacial binding strength: tape-peel test and pull-off test (Figure R1). As for tape-peel test, the interfacial binding strength is calculated by the peel plateau force divided by the sample width, in N/m. While the interfacial binding strength based on the pull-off test is calculated through the pulling force divided by contact area, in MPa. The units of measurement for two methods are different, so the interfacial binding strength cannot be directly compared. Here, we chose the tape-peel test method to measure the interfacial binding strength of our devices. This is because the used liquid glue (i.e. epoxy resin) in the pull-off test may diffuse into the sample to increase the contact area, leading to measurement errors of the pull-off test (Figure R1). Specifically, the T-peel test, a common method of tape-peel test, is employed in our work.

The performance comparison of our devices with those in the literature based on tape-peel tests is systematically illustrated in Table R1. The peel strength of our Au-RPU device is 1243.4 N/m, which is significantly higher than that of other devices based on inert metal functional layers, demonstrating the advantages of our DIC strategy.

In addition, the anti-friction experiments of Au-based devices with other substrates including Ecoflex elastomer, thermoplastic polyurethane (TPU) elastomer, and flexible polyimide (PI) were also conducted (Figure R2), further proving the excellent anti-friction performance of our Au-based stretchable devices.

The corresponding descriptions and **Supplementary Figure 9** and **Supplementary Table 2** have been added into the in the revised manuscript (Lines 138-140, Lines 147-151, Page 7; Page S9 and S30).

Figure R1. Schematic of measurement methods of the interfacial binding strength, including tape-peel test (90°-peel test and T-peel test) and pull-off test.

Supplementary Table 2 Comparison of the interfacial binding strength of our device with other reported works.

Device structure	Interface design	Test method	Peel strength (N/m)	Reference
Au/SEBS	Biphasic interpenetrating nanostructure	T-peel test	120 N/m	Nature 2023, 614, 456.

Au/PU fiber/PAM hydrogel	Mechanically interlocked interface	90° -peel test	< 60 N/m	Adv. Funct. Mater. 2020, 30, 1909540.
Au(Pt)-Ti/Polyimide	Chemical modification	T-peel test	< 400 N/m	Adv. Mater. Technol. 2021, 6, 2100149.
Pt-Ti/ Off-stoichiometry thiol-ene-epoxy film	Direct deposition	T-peel test	< 100 N/m	Biomaterials, 2023, 293: 121979.
Cu tape/Kapton film	Direct paste	T-peel test	~ 500 N/m	IEEE International Conference on Electro Information Technology, 2021, 409.
Au-RPU	Diffusion-induced cohesion	T-peel test	1243.4 N/m	This work
Au-WPU	Diffusion-induced cohesion	T-peel test	1017.6 N/m	This work

Figure R2. Anti-friction performance of Au-based samples with PDMS, SEBS, WPU, Ecoflex, TPU, PI and RPU substrates, respectively.

Q2: Anti-friction artificial skins also do not need to be limited to the use of Au, so the comparison and discussion with the other conducting materials should be made.

Response: Thank you for your comment. Here we carried out new experiments by using platinum (Pt), silver (Ag) and copper (Cu) as conductive layers to prepare stretchable devices. The comparison of electrical property and anti-friction performance of these devices are shown in **Supplementary Table 3**. In our work, two reference sheet resistance (R) values of 500 Ω/\square and 100 $M\Omega/\square$ are chosen as the criteria of conductive and insulating states, as the device's resistance is gradually increased during repeated rubbing. Similar to Au-RPU devices, the inert Pt-RPU device

can maintain conductive state after 49 ± 5 frictions, and falls into the insulating state after 64 ± 3 frictions, exhibiting strong electrical anti-friction ability. In other words, our approach is not limited to the use of Au, which can extend to other inert metals.

As to active Ag and Cu conducting materials, these materials are seldom employed to construct microcracks-based stretchable electronics. It is attributed that Ag and Cu microcracks/microfragments are easily oxidized by external environment, resulting in poor conductivity or even loss of conductivity¹. This phenomenon is further verified in our experiments. The initial resistances of the as-deposited Ag-RPU and Cu-RPU device are conductive, similar to inert Au- and Pt-based devices. After only 3 hours in the air, the Cu-RPU device would become non-conductive, due to the oxidation of Cu microcracks/microfragments exposed to the air. For Ag-RPU device, it would take about 1 month to become non-conductive in the air. Hence, these two active conducting materials are not suitable for microcracks-based stretchable electronics due to the easy oxidation of active microcracks/microfragments exposed to the air. As a consequence, our work focuses on the inert metal-based stretchable electronics, also seen in the “Introduction” section.

We chose Au, rather than Pt, as the functional layer for stretchable electronics because Au has a lower evaporation temperature, lower hardness and better ductility. We add the description and **Supplementary Table 3** in the revised manuscript (Lines 164-167, Page 8; Page S31).

Supplementary Table 3 Comparison of electrical property and anti-friction performance of stretchable devices with different conductive materials.

Device type	Stability in the air		Number of anti-friction (a.u.)	
	Conductivity after deposition	Conductivity in the air	Conductive state	Insulating state
Au-RPU	Conductive	Permanently conductive	53 ± 5	61 ± 4
Pt-RPU	Conductive	Permanently conductive	49 ± 5	64 ± 3
Ag-RPU	Conductive	Non-conductive after 1 month	\	\
Cu-RPU	Conductive	Non-conductive after 3 hours	\	\
Conductive state: $R < 500 \Omega/\square$ Insulating state: $R > 100 M\Omega/\square$				

Q3: How does the anti-friction performance of the Au-RPU device depend on the pattern and geometric parameters of the rough configuration?

Response: To investigate the effect of geometric parameters on anti-friction performance, different samples with varied surface roughness are fabricated. The polyurethane films with different surface roughness are obtained by regulating the urea

contents. As shown in Table R1, the anti-friction ability of the device increases from 36 ± 2 times on the smooth surface (Ra: 31 nm) to a maximum of 61 ± 4 times on the rough surface (Ra: 212 nm) with increasing the surface roughness. This result is attributed that the increase of roughness increases the effective contact area between Au grains and polyurethane substrate, which is verified by finite element analysis in the manuscript. However, the further increase of roughness will lead to the decrease of anti-friction performance. It could be explained by the fact that the excessive uneven rough surface makes the deposition of gold layer on polyurethane substrate prone to gaps and defects, and thus deteriorates the electrical anti-friction performance. In summary, the Au-RPU device with surface roughness of 211 nm exhibits the best anti-friction performance.

Table R1 Comparison of anti-friction performance of our device with varied surface roughness.

Device	Surface roughness (Ra)	Number of anti-friction (a.u.)	
		Conductive state	Insulating state
Au-WPU	31 nm	28 ± 3	36 ± 2
Au-RPU	103 nm	35 ± 2	41 ± 3
	168 nm	43 ± 3	50 ± 3
	211 nm	53 ± 5	61 ± 4
	430 nm	30 ± 2	35 ± 3
Conductive state: $R < 500 \Omega/\square$ Insulating state: $R > 100 M\Omega/\square$			

Q4: *How does the cyclic friction in Fig. 4b correlate to the use of the sensors in practical applications? Many previously reported sensors have already shown the robust performance of electrophysiological signal detections during daily use or even in extreme use cases. Therefore, it is important to provide the demonstrations/applications that can be challenging with the other reported sensors.*

Response: Thanks for the comments. As the discussion in the “Introduction” section, we focus on inert metal-based stretchable electronics in this work. The excellent anti-dynamic interference capability of our devices is a significant advantage over currently reported inert metal-based soft electrodes. Although these soft electrodes have achieved long-term, or even underwater stable bio-signal detection, the device usually experience relatively small interferences during the measurement. However, the device performance subjected to dynamic interferences like frictions is unclear, which requires

specific measurements and characterizations²⁻⁴. For instance, when the electrodes are attached to the skin or hairy scalp of the person, the strenuous exercise such as running and jumping can induce the external friction caused by the hair, skin and fabrics (**Supplementary Figure 19**). Therefore, it is vital to develop soft electrodes with dynamic anti-friction ability.

To clearly illustrate the superiority of our anti-friction electrodes, we here added two new demos that should be challenging with the other reported electrodes, as shown in **Supplementary Movie 3 and 4**. One demo was to attach Au-RPU and Au-PDMS electrodes to the chest skin and compare the recording quality of ECG signals obtained before and after dynamic rubbing (**Supplementary Movie 3**). In the second demo, Au-RPU and Au-PDMS electrodes were attached to the hairy scalp and the obtained EEG signals were compared before and after rubbing against scalp hair (**Supplementary Movie 4**). Au-RPU electrodes can record high-quality physiological electrical signals even after repeated frictions with external fabrics and hair. In contrast, the Au functional layer of Au-PDMS electrodes is easily destroyed by dynamic frictions, resulting in device failure. These comparative experiments confirm the excellent tolerance of our Au-RPU devices to dynamic interference, which is an apparent superiority over conventional reported inert metal-based stretchable electrodes.

We add Figure 4, Supplementary Figure 19, Movie 3 and 4, and descriptions in the revised manuscript (*Lines 268-281, Page 13; Page S21*).

Supplementary Figure 19 Image of (a) ECG and (b) EEG signal detection by Au-RPU electrodes.

Q5: *The validation and application of the EEG and EMG signals can be helpful to include.*

Response: Thank you for the suggestion. We have added two new demos about ECG and EEG recording in the revised manuscript (also seen in the above response to Q4). The aforementioned two demos have well demonstrated the practicability and advantages of our anti-friction devices against the other reported electrodes. A detailed discussion of stable SNR value of EMG recording is further added in the revised manuscript.

Q6: *The Au-RPU electrode exhibits much lower contact impedance than that of the commercial gel electrodes, but gives a signal-to-noise ratio lower than many other*

published results (that do not even show superior contact impedance). Could the authors comment and explain?

Response: Thank you for your comment. For the calculation of SNR, there are different methods reported in previous literatures, including power-based SNR, energy-based SNR, amplitude-based SNR, peak SNR and statistical SNR⁴⁻⁷. Considering the calculation difference, we did not directly compare the SNR value of our device with previously reported results (because most of reports do not give the specific calculation processes in detail). Here, we compared the SNR values by using the same calculation method. We obtained the SNR values of EMG signals by using an algorithm that was reported in the Ref 8⁸. This algorithm is a statistical approach without the need of preprocessing EMG signals in the time domain (the calculated values in some reports are obtained from the signal data after preprocessing). A high-fidelity EMG signal in the Figure 3 of Ref 8 was used as a benchmark, as shown in Figure R3(b). The SNR value of the EMG signals in the Figure R3(b) was calculated to be 20 dB using this method. In our work, the SNR value of the EMG signals (Figure R3(a)) is 25.7 ± 0.8 dB by employing the same calculation method, which is comparable to the reported value in the Ref 8. This result indicates that the anti-friction electrodes used in our work also have a high SNR.

Figure R3. EMG signals collected by our Au-RPU device (a) and extracted from Ref 8 (b), and the comparison of their SNR values (c).

Q7: *The comparison of the fibrous pressure sensors in Table S1 is quite limited. Please benchmark the complete performance parameters (e.g., sensitivity, linearity, the limit and range of detection, and long-term stability, among others) of the sensor against those in the literature to clearly demonstrate the novelty. The comparison can also go beyond the fibrous sensors to include the other pressure sensors.*

Response: We really appreciate the reviewer's valuable suggestion. In the revised manuscript, we carried out a comprehensive performance comparison between our Au-based pressure sensor array and other reported pressure sensors, as shown in **Supplementary Table 8**. Since our work is relevant to metal electrode-based pressure sensors, we here emphasize the comparison of our sensor array among the relevant metal-electrode-based sensors. Although many pressure sensors have been implemented to detect force/pressure with high sensitivity, it is still difficult for them to accurately and reliably detect concentrated large forces/pressures (more than 1 MPa) on sharp three-dimensional (3D) surfaces because of friction interference.

In this work, we presented an anti-friction crossbar-structured pressure sensor array

based on inert Au bottom and top electrodes. Due to the excellent tolerance to interfacial friction of the Au-RPU electrodes, this array can be conformably attached to a person's palm to reliably perceive concentrated large pressures up to 22.2 MPa with a crosstalk-free readout circuit when grasping objects with sharp surfaces. In a word, our Au-based pressure sensor array exhibits better comprehensive performance than previously reported metal electrode-based pressure sensors, as shown **Supplementary Table 8**. Supplementary Table 8 and relevant descriptions are added in the revised manuscript (*Lines 308-328, Page 15; Lines 329-351, Page 16; Lines 352-354, Line 373, Page 17; Lines 374-375, Page 18; Pages S22-28; Page S37*).

Supplementary Table 8 Comparison of our pressure sensor array with the metal-electrode-based pressure sensors.

Device electrode	Electrode stretchability	Sensing performance			Anti-friction ability	Reference
		Sensitivity	Detection range	Long-term stability		
Ag nanowire electrode	No	$\Delta I/I_0 = 17$, $0 \leq P \leq 1$ kPa; $\Delta I/I_0 = 10$, $2 \leq P \leq 10$ kPa	0-10 kPa	Yes	N/A	ACS Nano 2022, 16, 368.
Ag nanoparticle electrode	No	$\Delta I/I_0 > 100$, $0 \leq P \leq 100$ kPa	0.05-900 kPa	Yes	N/A	Adv. Mater. Technol. 2022, 7, 2100428.
Ag nanoparticle electrode	No	$\Delta I/I_0 \approx 50$, $0 \leq P \leq 50$ kPa	5-600 kPa	Yes	N/A	Adv. Mater. Interfaces 2022, 9, 2200621.
Ag paste interdigital electrode	No	$\Delta I/I_0 \approx 70$, $0 \leq P \leq 15$ kPa	0-667 kPa	Yes	N/A	Smart Mater. Struct. 2019, 28 105027.
Cu foil electrode	Yes	$\Delta R/R_0 \approx 60\%$, $0 \leq P \leq 1$ kPa	0-20 kPa	Yes	N/A	Composites Part B 2021, 225, 109243.
Cu wire electrode	No	$\Delta I/I_0 \approx 0.2$, $0 \leq P \leq 1$ kPa	0-20 kPa	Yes	N/A	J. Mater. Chem. C 2020, 8, 16774
Cr/Au electrode	No	$\Delta R/R_0 \approx 0.25\%$, $0 \leq P \leq 30$ kPa	0-30 kPa	Yes	N/A	Nat. Biomed. Eng.

						2020, 4, 997.
Cr/Au bottle electrode CNT top electrode	Yes	$\Delta R/R_0 \approx 99\%$, $0 \leq P \leq 2.5$ kPa; $\Delta R/R_0 \approx 90\%$, 2.5 kPa < $P \leq 90$ kPa	0-90 kPa	Yes	N/A	Science 2018, 360, 998.
Cr/Au electrode	No	$\Delta I/I_0 \approx 0.8$, $0 \leq P \leq 0.15$ kPa	0-50 kPa	N/A	N/A	Sensor. Actuat. A 2018, 280, 261.
Ti/Au electrode	No	$\Delta C/C_0 \approx 7$, 10.1 dBar $\leq P \leq 10.9$ dBar	10.1-10.9 dBar	N/A	N/A	npj Flex. Electron. 2018, 2, 13.
Au electrode	Yes	0.141 kPa ⁻¹ , < 1 kPa; 0.01 kPa ⁻¹ , > 10 kPa 100 kPa pressing.	0-100 kPa	Yes	Anti-friction ability by polyurethane encapsulation layer	Science 2020, 370, 966.
Au electrode	Yes	$\Delta R/R_0 \approx 40\%$, $0 \leq P \leq 3.1$ MPa; $\Delta R/R_0 \approx 33\%$, 3.6 MPa $\leq P \leq 22.2$ MPa	0-22.2 MPa	Yes	Excellent anti-friction ability due to DIC strategy	This work

Q8: *The use of the anti-friction property is not clearly outlined in the pressure/force sensing, as many previous reports have shown reliable sensing "after 1000 pressing deformations".*

Response: We appreciate the reviewer's constructive comments. Although existing inert Au electrode-based pressure sensors have shown long-term stability (after many cycles), the real deformation between sensor unit and object during the measurement is usually small, as summarized in **Supplementary Table 8**. In these scenarios, the sensors only experience limited external pressures (usually less than 100 kPa), which fails to the requirements for anti-friction electronics^{9, 10}. Once experiencing a large external pressure during the measurement, these sensors are prone to be deteriorated, due to the weak interfacial binding strength between inert metals and elastic substrates. Our work can solve the interface issue between inert metals and elastic substrates towards stretchable anti-friction electronics.

To clarify this issue more clearly, we supplemented a new comparison experiment. As a contrast, a new pressure sensor array was constructed based on patterned Au-PDMS electrodes using the same assembly process. The device could not normally work under large external forces. This is because most of the patterned Au layers on

the PDMS substrate would be scratched off when the array is subjected to a small pressing force and small deformation (even after assembly process) due to the poor interfacial anti-friction ability (**Supplementary Movie 10** in the revised manuscript). In contrast, our device can be attached to sharp 3D surfaces, and reliably perceive the force signals up to 22.2 MPa even when subjected to concentrated deformation stresses, demonstrating the excellent anti-friction ability of our electrodes fabricated by the DIC strategy.

To further demonstrate the unique advantages of our work, we have supplemented new demonstrations, which are shown in the revised **Figure 5** and **Supplementary Movie 7-9**. Pressure sensor array based on our anti-friction Au-RPU electrodes was attached to the hand palm to measure the strength of the grip when grasping objects with different surface sharpness. Notably, our device can reliably record the magnitude and distribution of force/pressure when grasping a horned melon with spinous surface (concentrated deformation), which is attributed to the excellent anti-friction ability of our Au-RPU electrodes (**Supplementary Movie 8, 9**). Meanwhile, our device can be conformably attached to the smooth surface of an apple during the grasping process to exhibit a more uniformly varied pressure distribution (**Supplementary Movie 7**). In contrast, Au-PDMS electrodes-based pressure sensor array could not work. Figures (**Figure 5, Supplementary Figure 20-24**), **Supplementary Movie 7-10** and descriptions are added in manuscript (*Lines 329-354, Pages 15-17; Lines 374-375, Page 18; Pages S22-28; Page S37*).

Figure 5 Anti-friction APSA. **a**, Schematic illustration of a bionic robot palm integrated with tactile perception to grasp an object. **b**, Schematic of structure and location of the APSA with 36 sensors (3 rows \times 12 columns) and 15 electrodes. **c**, Response of the sensing unit of APSA before and after 1000 frictions. **d**, Optical image of the APSA attached on the palm to grasp an apple with smooth surface. **e**, Zoom-in image showing the conformable contact between grasper and the apple. **f**, Pressure force maps of the APSA when grasping the apple. **g**, Optical image of the APSA attached on the palm to grasp a horned melon with spinous surface. **h**, Zoom-in image showing unconformable contact between grasper and the horned melon with sharp deformation domains. **i**, Pressure force maps of the APSA when grasping the horned melon. **j**, Stability of the APSA during cyclic sharp deformation tests under 2.2 MPa pressure. **k**, SEM images of Au-RPU electrodes used for the APSA after the first (left) and 200 times (right) sharp deformation test under 2.2 MPa pressure. Scale bar: 5 μ m.

Q9: How does the sensor performance change with variations in environmental conditions (e.g., temperature and highly humid or water moisture environments) or with mechanical deformations (e.g., stretching, bending, or twisting)?

Response: We evaluate the effects of ambient temperature, humidity, mechanical stretching and bending on the sensor performance, which is shown in **Supplementary Figure 22**. Compared with the original state, the sensor performance remains relatively stable without notable variations even at high temperature (up to 60 °C), high humidity (relatively humidity: 100%), stretching (20% strain) or bending condition (90° bending). **Supplementary Figure 22** and descriptions are added in manuscript (*Lines 324-327, Pages 15; Page S24*).

Supplementary Figure 22 Response of the pressure sensor during the cyclic loading/unloading process at 2.2 MPa pressure with various environmental conditions, including the original, 40 °C, 50 °C, 60 °C, high humidity, stretching and bending state.

Q10: *The introduction section may also be updated to include the state-of-the-art literature on fabrication methods to prepare deformable sensors or functional circuits on 3D freeform surfaces, multimodal sensors with decoupled sensing mechanisms, and standalone stretchable sensing platforms to provide comprehensive background and future opportunities.*

Response: Thanks for the suggestion. We have added the state-of-the-art research works and the relevant discussions in the introduction section (*Lines 49-53, Line 65, Page 3; Lines 71, 72, 74, 77, Page 4*). The latest advances in the fields of conformable/stretchable electrodes, multimodal sensors, functional circuits, standalone wearable sensing platforms and their fabrication methods have been supplemented.

Reviewer #2:

Comments: *The method involves using hydrophilic PU to coat the gold grains, creating strong hydrogen bonding and resulting in a high interfacial binding strength. Through an examination of the friction resistance of the stretchable electrodes, the authors have demonstrated the feasibility of detecting human bio-signals and achieving circuit interconnects. However, it is worth noting that while the stretchable electrodes exhibit increased mechanical resistance against external rubbing forces, their mechanical stretchability remains relatively poor. This limitation could potentially restrict its applications in soft electronics. Additionally, the manuscript lacks dedicated in-depth analysis and demonstrations, which are necessary for publication in a prestigious journal like Nature Communications. Therefore, I do not support the publication of this paper in its current form. Below are detailed comments addressing specific aspects of the manuscript.*

Response: We really appreciate the reviewer's constructive comments. From the newly added tests on mechanical property, both pure WPU and Au-WPU films in our work exhibit high stretchability of ~ 900% that is comparable with widely used PDMS- and SEBS-based stretchable electronics^{2, 11} (Table R2). As to our Au-RPU electrode, its electrical conductive mechanism upon stretching is attributed to the metallic microcrack conductive paths^{12, 13}. Our Au-RPU electrode can maintain microcrack conductive paths at 400% tensile strain, which is almost the highest stretchability among all microcracks-based stretchable electrodes. The detailed discussion and comparison of electrical performance of microcracks-based stretchable electronics can be seen in the response to the following Q1 and Q2.

In addition, we have addressed the reviewer's concerns about the dedicated in-depth analysis and demonstrations through additional data as below.

1. We have carried out *in-situ* temperature-dependent FTIR to identify the hydrogen bonding interaction in WPU in detail, which is useful to understand and validate the DIC strategy. Meanwhile, we have added the comparisons and discussions on interfacial binding strength of metal-based electronic devices (**Supplementary Table 2**), electrical properties and anti-friction abilities of Au-based stretchable electronics (**Supplementary Table 3 and 5**), and comprehensive performances of metal electrode-based pressure sensors (**Supplementary Table 8**), which are used to verify the superior performance of our Au-RPU-based devices.

2. We have added three demos to leverage the unique advantage of our anti-friction Au-RPU device prepared by DIC strategy over other strategies. The first two demos are to attach the Au-RPU electrodes and Au-PDMS electrodes to the chest skin or the hairy scalp to compare their electrophysiological signals before and after dynamic rubbing (**Figure 4 and Supplementary Movie 3 and 4** in the revised manuscript). In the third demo, a newly assembled pressure sensor array based on our anti-friction Au-RPU electrodes can be attached to sharp 3D surfaces and reliably perceive the force signals up to 22.2 MPa even when subjected to concentrated deformation stresses (**Figure 5 and Supplementary Movie 7-9** in the revised manuscript).

Table R2 Mechanical property of pure WPU and Au-WPU film.

	Stretchability (%)	Tensile strength (MPa)	Young modulus (MPa)
Pure WPU	914	0.76	1.28
Au-WPU	903	0.75	1.31

Q1: *The primary focus of this research is to enhance the mechanical integrity of gold-based stretchable interconnects, particularly developing the electrodes more durable to external stimuli (i.e., tangentially rubbing under a given force). Despite achieving remarkable improvements in the adhesion properties of the Au thin film via increasing the interfacial binding strength, the authors have not been able to maintain the electrical conductivity of the interconnects when subjected to mechanical strain, which is a crucial requirement for their development in more practical applications. In specific, the authors claim that the Au-RPU device exhibits a low resistivity ($1.09 \times 10^{-3} \Omega m$) even at a large stretchability of 400%. Not only the substantial increase of (approximately fourfold) the resistivity upon 400% mechanical stretching is expected to hinder the applicability of the stretchable interconnects but also a relatively low friction number (max of 1022) achieved in this study is considered insufficient performance compared to previously reported research findings. Furthermore, the authors did not conduct comprehensive stretchability tests.*

Response: Thank you for your comments. As described by the reviewer, this work focuses on anti-friction stretchable electronics based on non-stretchable Au functional layer due to the remarkable advantages of inert metals (as discussed in the response to Q2 of reviewer #1 and the “Introduction” section). For such devices, the non-stretchable Au layers will be fractured into microfragments with branched microcracks induced by the strain concentration upon stretching^{12, 13} (Figure R4). These Au microfragments will form conductive percolating networks when the elastic substrate is stretched^{14, 15}, leading to Au microcracks-structured stretchable electronics. Normally, it is inevitable that the electrical resistivity of the device increases as the tensile strain increases for the microcracks-based stretchable electrodes, which is consistent with the previous reports^{2, 16}. We have compared the electrical performance of our device with other Au microcracks-based stretchable devices, as shown in **Supplementary Table 5** (also shown in the response to Q2 below). The Au layer of most microcracks-based stretchable electrodes has been disconnected and lost their conductivity at 200% tensile strain, while our Au-RPU device can retain good conductivity (roughly 1×10^3 S/m) at 400% strain due to the strong interfacial binding strength of our Au-RPU device through the unique DIC approach. This result demonstrates that our Au-RPU stretchable electrodes is superior to other microcracks-based stretchable electrodes. Although we highly agree your view that stretchable interconnects/electrodes are better to maintain the constant electrical conductivity when subjected to mechanical strain for real applications, it is difficult for such kind of stretchable interconnects/electrodes due to the intrinsic microcrack mechanism.

In this work, the value of 1022 frictions was collected by direct contact with Au layer

and under a vertical load of 10 N (130 kPa). Our test condition is much harsher than the reported work with encapsulation layer and small load of 0.5 N (**Supplementary Table 5** shown in the response to the following Q2). In fact, the anti-friction number of our device can be significantly improved by using a relatively low load value. Furthermore, we compared the interfacial binding strength (**Supplementary Table 2**) and added anti-friction experiments of Au-based devices with other typical substrates (Figure R2) to evaluate the anti-friction ability of our Au-RPU device. The details are shown in the response to Q1 of reviewer #1. The comprehensive comparisons verify the excellent electrical tolerance of our Au-based stretchable device against external friction interferences.

According to the reviewer's suggestion, we have supplemented the comprehensive stretchability experiments to evaluate the device stability and long-term durability (**Supplementary Figure 12 and 13**). Our Au-RPU device exhibits good stability when applying five successive loading and unloading cycles at peak strains of 25%, 50% and 100% (**Supplementary Figure 12**). Furthermore, the resistance response of the device at a peak strain of 100% over 1,000 cycles is stable, demonstrating the long-term durability of our device (**Supplementary Figure 13**). Supplementary Figure 12, 13 and descriptions are added in the revised manuscript (*Lines 193-197, Page 9; Pages S14-15*).

Figure R4 SEM image of microcracks-based Au-RPU device upon tensile strain.

Supplementary Figure 12 Stability of the Au-RPU device against cyclic tensile deformation at 25%, 50% and 100% strain.

Supplementary Figure 13 Stability of the Au-RPU device during 1,000 cycles of the stretch/release process at 100% tensile strain. Inset shows the 10 cycles of the stretch/release process from 701 to 710 cycle

Q2: *The authors primarily focused on comparing the performance and mechanical durability of the stretchable interconnects with PDMS and SBS-based stretchable electrodes. However, it is evident that the Au-RPU interconnects would undoubtedly exhibit superior electrical and mechanical properties due to their enhanced interfacial binding strength. To establish the applicability of stretchable interconnects in a wide range of soft electronics, it would be more reasonable to compare them to previously successful demonstrations of gold-based stretchable interconnects, such as those published in Nature (500, 59-63, 2013) and Nature Nanotechnology (12, 907-913, 2017). This approach would provide a more solid basis for claiming the excellence of the anti-friction gold-based stretchable electrodes.*

Response: According to your constructive suggestions, we have benchmarked our device with other reported gold-based stretchable devices, as shown in **Supplementary Table 5**. The gold layer of most gold-based stretchable devices has been disconnected and lost conductivity at 200% tensile strain, while our Au-RPU device exhibits

excellent conductivity at 400% strain. More importantly, the electrical stability of those reported gold-based stretchable devices is not provided when subjected to external friction interference, while our device can maintain highly conductive state upon more than 1000 frictions. These results highlight the excellent electrical property and anti-friction ability of our Au-RPU device. Supplementary Table 5 and descriptions are added in the revised manuscript (*Lines 179-182, Page 9; Pages S33-34*).

Supplementary Table 5 Comparison of electrical properties and anti-friction ability of gold-based stretchable electronics.

Device	Conductive mechanism	Electrical property upon strain	Anti-friction ability	Reference
Au-PU	Layer-by-layer assembly of Au	1×10^6 S/m, 0% strain; Break, 120% strain	N/A	Nature 2013, 500, 59-63.
	Au-PU blending	1×10^5 S/m, 0% strain; 5×10^3 S/m, 400% strain		
Au-PVA nanomesh	Microcracks	3×10^{-3} S, 0% strain; Break, ~50% strain	N/A	Nat. Nanotech. 2017, 12, 907.
Au-Supramolecular polymeric materials	Microcracks	Low resistance, 0% strain; Break, ~400% strain	N/A	J. Am. Chem. Soc. 2018, 140, 5280.
Au-Parylene-Polyurethane	Microcracks	55 Ω , 0% strain; Break, ~40% strain	N/A	Nat. Nanotech. 2019, 14, 156.
Au-PDMS	Microcracks	25-300 Ω , 0% strain; Break, ~70% strain	N/A	Adv. Mater. 2016, 28, 6359.
Au-PDMS meshed film	Microcracks	12 Ω /sq, 0% strain; Break, 94% strain	N/A	ACS Sens. 2020, 5, 3165.
Au-Shape memory polymer	Microcracks	85 Ω /sq, 0% strain; Break, ~200% strain	N/A	Appl. Phys. Lett. 2016; 108, 061901.
Au-PDMS	Microcracks	500 Ω , 0% strain; Break, ~120% strain	N/A	J. Appl. Phys. 2019, 125, 165305.
Au-PDMS	Microcracks	~20 Ω , 0% strain; ~1 k Ω , 300% strain; Break strain, not given	N/A	Nat. Electron. 2022, 5, 784.

Au-PVA-PU	Microcracks	Pressure sensor: $\Delta C/C_0 = 1.4$, 100 kPa pressing.	Force: 0.5 N; Encapsulation: polyurethane layer; Performance: remain stable at provided 300 frictions.	Science 2020, 370, 966.
Au-RPU	Microcracks	$\sim 3 \times 10^6$ S/m, 0% strain; $\sim 1 \times 10^3$ S/m, 400% strain	Force: 10 N (130 kPa); No encapsulation; Performance: Remain conductivity after 1022 ± 76 frictions.	This work

Q3: *It is considered oversimplified to merely demonstrate human bio-signals and the electrical response to a certain level of impact as a measure of the development of stretchable electrodes. Specifically, the manuscript lacks the presentation of an electrical circuit setup that enables real-time monitoring of signal changes in response to mechanical impact. It is crucial that this construction of the electrical circuit be shown in a video or supplementary material to provide a more comprehensive understanding of the research.*

Response: As the reviewer's suggestion, we have supplemented the detailed procedure of electrical testing of human bio-signals in "Methods" section (*Lines 412-416, Page 19-20*), **Supplementary Figure 19 and Movie 2**. Briefly, the test was performed using the commercial readout circuit connected to a software on notebook. In addition, the section about the circuit setup of the pressure sensor array is reorganized in the original Methods part "Fabrication and readout of anti-friction APSA", **Supplementary Figure 20 and 23** (*Lines 422-447, Page 20-21; Lines 499-506, Page 23-24; Page S22; Page S25*). Thanks for your careful work.

Supplementary Figure 19 Optical images of (a) ECG and (b) EEG detection.

Supplementary Figure 20 Optical image of our 3×12 pressure sensor array based on Au-RPU electrode and the signal readout hardware circuit.

Q4: *It is necessary to express all electrical performance measurements in terms of sheet resistance (ohm/sq.). Additionally, the normal loads should be presented as pressure values based on the Hertzian contact model. This approach is crucial for facilitating clear comparisons and ensuring a comprehensive understanding of the level of pressure applied in the experiments.*

Response: Thanks for your suggestion. We have modified the resistance into the sheet resistance, and changed the load into the pressure value based on the Hertzian contact model. All the Figures and descriptions are modified in the revised manuscript (Line 35, Page 2; Lines 89-91, Page 5; Lines 120-121, Page 6; Lines 150-152, Page 7; Line 168, Page 8; Line 314, Lines 321-322, Page 15; Line 368, Page 17).

Reviewer #3:

Comments: *The manuscript entitled "Anti-friction gold-based stretchable electronics enabled by interfacial diffusion-induced cohesion" submitted by Ming Liu and coworkers presented a chemical modification-free diffusion-induced cohesion (DIC) strategy to achieve strong interfacial binding between inert Au and waterborne polyurethane (WPU) for anti-friction electronics. While the material design presented in this work is intriguing, there are several unsupported claims regarding the anti-friction ability of Au-RPU throughout the figures and text. 1) Given the potential influence of temperature and humidity on the functional groups and molecular interactions within the composite materials, the authors should provide a range of conditions that ensure consistent performance. 2) In addition, the mechanical mismatch between solid inert metals and elastomers significantly affects interfacial defects; however, the modulus of the materials is not addressed in the manuscript. Because of these two reasons, the impact of this work may be limited. Overall, I recommend the addition of more experiments and appropriate discussion for the aforementioned issues. Here are some detailed comments.*

Response: We thank the reviewer for the positive comments. According to the reviewer's advice, *in-situ* FTIR spectra have been carried out to investigate the influence of temperature on the hydrogen bonding interaction, as shown in **Figure 3g and h**. The results reveal the maintenance of the hydrogen bonding network during the temperature changes (detailed discussion is shown in the revised manuscript). In addition, we evaluate the sensor performance of the Au-based pressure sensor upon the high-humidity environment, as discussed in the response to Q9 of Reviewer #1 and **Supplementary Figure 22**. The sensor performance remains consistent with the original state without notable variations.

Meanwhile, the Young's modulus for different flexible or stretchable substrates have been added in **Supplementary Table 1**. It can be concluded that our WPU has lower Young's modulus than PDMS, SEBS, TPU, PET and PI, which contributes to the conformability and stretchability of the device.

At last, in order to support claims regarding to the anti-friction ability of our Au-RPU devices, we have added additional data that are shown below.

1. We have added comparison and discussion on interfacial binding strength of metal-based electronic devices (**Supplementary Table 2**), electrical properties and anti-friction abilities of Au-based stretchable electronics (**Supplementary Table 3 and 5**), and comprehensive performances of metal electrode-based pressure sensors (**Supplementary Table 8**), to verify the superior performance of our Au-RPU-based devices.

2. We have added three demos and descriptions to leverage the unique advantage of our anti-friction Au-RPU device prepared by DIC strategy over other strategies. The first two demos are to attach the Au-RPU electrodes and Au-PDMS electrodes to the chest skin or the hairy scalp to compare their electrophysiological signals before and after dynamic rubbing (**Figure 4 and Supplementary Movie 3 and 4** in the revised manuscript). In the third demo, a newly assembled pressure sensor array based on our anti-friction Au-RPU electrodes can be attached to sharp 3D surfaces and reliably

perceive the force signals up to 22.2 MPa even when subjected to concentrated deformation stresses (**Figure 5 and Supplementary Movie 7-9** in the revised manuscript).

Supplementary Table 1 Young's modulus of different flexible substrate materials.

	WPU	PDMS	SEBS	Ecoflex	TPU	PET	PI
Young's modulus (MPa)	1.28	2.16	5.77	0.47	25.15	2102.82	805.40

Q1: The authors mentioned the "cohesion effect of adjacent wetted Au grains in the interfacial layer between Au and PU." While I understand the author's intention behind this concept, the definitions of molecular interactions and hydrogen bonds occurring between Au and PU are not clear. Additionally, the exact molecular structures are not provided, making it challenging to comprehend the notion of "molecular glues." If the authors could provide a more explicit representation or description schematic of the molecular structure involved in the Au-PU interface, it will help readers to better understand this work.

Response: Thanks for the reviewer's valuable suggestion. To make this work easier to understand, we added the molecular structure unit of WPU, the schematic diagram of oxygen-containing groups and the hydrogen bonding interactions in the revised manuscript. As shown in **Supplementary Figure 1**, it can be deduced from the WPU molecular structure where the oxygen-containing groups are consist of hydroxyl groups, carboxyl groups and ester groups. These groups, together with the water molecules (contain hydroxyl groups), act as "molecular glues" to connect the wetted Au and WPU at the interface because they are able to construct hydrogen bond cohesion effect. Supplementary Figure 1 and description is added in the manuscript (Lines 110-111, Line 113, Page 6; Line 249, Page 12; Page S3).

Supplementary Figure 1 Molecular structure unit of WPU. Schematic illustration of molecular structures of molecular glues and their hydrogen bonding interactions in the Au-RPU interface.

Q2: *In Figure 1c, the authors measured the peel strength as a function of distance for Au-WPU and Au-RPU, and these values fluctuate significantly. Given the envisaged use of these sensors in scenarios involving bending or movement, such fluctuations cause undesirable noise, thereby reducing the signal-to-noise ratio. Consequently, it is imperative for the authors to provide a comprehensive discussion to mitigate these concerns and offer suitable solutions.*

Response: The curve fluctuation in T-peel tests should be caused by several factors such as the non-uniformity of binding material and substrate, interface effect, and instrumental noise. The different mechanical properties and structural characteristics of the binding material and the substrate may lead to uneven stress distribution during the peeling process. Meanwhile, the interface binding processing deviation of the adhesive interface can also cause the fluctuation of the curve. Because of those unavoidable error factors, the curve fluctuation in T-peel test has been commonly observed in previously reported works^{11, 17, 18}.

Notably, a feasible strategy to mitigate the fluctuation is to read the values of the peeling force plateaus to characterize the binding properties of the sample, which has been widely adopted by researchers. In addition, filtering algorithm or average value calculation can also be used to smooth the test data and reduce the fluctuation of the curve. These relevant discussions and potential solutions have added in the revised manuscript (Lines 136-138, Page 7).

Q3: *In Figure 1, The authors provided XPS results for polyurethane (PU). While the authors provided the XPS result of PDMS, there are no C 1s peaks or Au 4f peaks, or any SEBS peaks that can be compared with those of PU. In addition, it may be inaccurate to identify the molecular interaction between Au, Pu, and water solely through the increased intensity observed in the XPS results. To strengthen the author's claim, it would be valuable to include additional evidence and analysis. In my opinion, exploring alternative approaches or techniques that can offer more direct information of hydrogen bonds, would be beneficial. The provided reference could provide insight into analyzing peak shift in XPS in terms of hydrogen bonding.*

Ref: Lei, Y., Tang, Z., Zhu, L., Guo, B., & Jia, D. (2011). Functional thiol ionic liquids as novel interfacial modifiers in SBR/HNTs composites. Polymer, 52(5), 1337-1344.

Response: According to the reviewer's constructive advice, *in-situ* FTIR spectra have been conducted to directly demonstrate the hydrogen bonding interaction in WPU. The relevant discussion and references have been added in the revised manuscript (Lines 232-246, Page 11-12; Lines 460-465, Page 21-22). Compared with XPS test, *in-situ* temperature-dependent FTIR can observe the evolution of molecular hydrogen bonding interaction in an intuitive way, as shown in **Figure 3g and h**. Relevant experimental data, discussion and references have been added in the revised manuscript: "*The hydrogen bonds were verified by in-situ Fourier transform infrared spectroscopy (FTIR;*

Figure 3g, h; Supplementary Figure 17). Two stretching vibration bands at 3385 and 3316 cm^{-1} represent the ordered hydrogen-bonding (H-bond) –OH and –NH– group, respectively. These bands move to a higher wavenumber upon heating, accompanied by the intensity reduction. These results indicate that the ordered H-bond –OH and –NH– groups in WPU are transformed into disordered H-bond types (Fig. 3g). The C=O wide peak at 1716 cm^{-1} is combined by the bands of free (1730 cm^{-1}), disordered H-bond (1716 cm^{-1}) and ordered H-bond C=O (1692 cm^{-1}). When the temperature is increased, the peak moves toward higher wavenumber with band intensity increasing for the free C=O and decreasing for the H-bond C=O groups, revealing the presence of hydrogen bonding in WPU. Notably, the peak intensity of disordered H-bond C=O is relatively stable with the slightly tendency of up-going first and then going down corresponding to the rearrangement of hydrogen bonds, which would be responsible to maintain the hydrogen bonding network.”

Figure 3g and h. Temperature-dependent FTIR spectra of WPU upon heating from 30 °C to 100 °C: (g) 3500–3150 cm^{-1} ; (h) 1800–1620 cm^{-1} .

Q4: In Figure 4c, the impedance value of the friction sample is higher than the original in the low-frequency region, but lower in the high-frequency region. It is required to define the impedance relaxation behavior according to frequency and provide an explanation for the observed differences in impedance at low (< 10 Hz) and high ($> 10^5$ Hz) frequency regions.

Response: Thank you for your comment. To further validate this phenomenon, we repeated the impedance test of different samples before and after frictions, as shown in Figure R6. All samples before and after frictions exhibits similar frequency-dependent interfacial contact impedance behaviors. With the increase of the frequency, the interfacial contact impedance will decrease, which is consistent with previous reports^{4, 19}. At low frequency range, the impedance relaxation behavior is dominated by processes with longer time constants. Typically, these processes include charge migration at the electrode and ion diffusion in the electrolyte²⁰. As the frequency increases, the relaxation behavior is translated into a fast process with shorter time constants such as interfacial charging or molecular polarization²¹⁻²³. It should be noted that all samples before and after frictions exhibit much lower interfacial impedance than

that of commercial Ag/AgCl electrode, which guarantees the signal fidelity in electrophysiological measurement.

As the reviewer mentioned, the impedance of the three samples after friction is higher than the original impedance before friction at low frequency regions (from 1 Hz to around 10^2 Hz). This is because friction causes the resistance value of the device to increase slightly, making the charge migration slower. At the high frequency region, the impedance of the three samples after friction is very close to the original impedance before friction (although there is a slight decrease). In this region, the resistance increase effect due to the friction would not dominates the change of interfacial impedance, and the slight impedance decrease may be related to the charging and polarization state of the sample surface after frictions^{20, 21}. All explanations and discussions about the impedance relaxation behavior and the impedance differences at low and high frequency regions have added in the revised manuscript (*Lines 295-297, Page 14*).

Figure R6. Electrical interfacial contact impedance measurements of commercial Ag/AgCl electrodes and three samples of Au-RPU electrodes before and after 1000 frictions.

Q5: In Figure 3a, there is a typographical error in the y-axis label. The word "Aomic" should be corrected to "Atomic."

Response: Thanks for the careful reading. We have revised this typographical error in Figure 3a.

References

1. Hyder A, *et al.* Identification of heavy metal ions from aqueous environment through gold, Silver and Copper Nanoparticles: An excellent colorimetric approach. *Environmental Research* **205**, 112475 (2022).
2. Jiang Z, *et al.* A 1.3-micrometre-thick elastic conductor for seamless on-skin and implantable sensors. *Nat. Electron.* **5**, 784-793 (2022).
3. Ji S, *et al.* Water-Resistant Conformal Hybrid Electrodes for Aquatic Endurable Electrocardiographic Monitoring. *Adv. Mater.* **32**, 2001496 (2020).
4. He K, *et al.* An On-Skin Electrode with Anti-Epidermal-Surface-Lipid Function Based on a Zwitterionic Polymer Brush. *Adv. Mater.* **32**, 2001130 (2020).
5. Liu Y, Cheng Y, Shi L, Wang R, Sun J. Breathable, Self-Adhesive Dry Electrodes for Stable Electrophysiological Signal Monitoring During Exercise. *ACS Appl. Mater. Inter.* **14**, 12812-12823 (2022).
6. Widmann A, Schröger E, Maess B. Digital filter design for electrophysiological data – a practical approach. *Journal of Neuroscience Methods* **250**, 34-46 (2015).
7. Suarez-Perez A, *et al.* Quantification of Signal-to-Noise Ratio in Cerebral Cortex Recordings Using Flexible MEAs With Co-localized Platinum Black, Carbon Nanotubes, and Gold Electrodes. *Front. Neurosci-switz* **12**, 862 (2018).
8. Agostini V, Knaflitz M. An algorithm for the estimation of the signal-to-noise ratio in surface myoelectric signals generated during cyclic movements. *IEEE Trans. Biomed. Eng.* **59**, 219-225 (2012).
9. Lee S, *et al.* Nanomesh pressure sensor for monitoring finger manipulation without sensory interference. *Science* **370**, 966-970 (2020).
10. Lee S, *et al.* Ultrasoft electronics to monitor dynamically pulsing cardiomyocytes. *Nat. Nanotechnol.* **14**, 156-160 (2019).
11. Jiang Y, *et al.* A universal interface for plug-and-play assembly of stretchable devices. *Nature* **614**, 456-462 (2023).
12. Liu Z, *et al.* High-adhesion stretchable electrodes based on nanopile interlocking. *Adv. Mater.* **29**, 1603382 (2017).
13. Huang S, Liu Y, Zhao Y, Ren Z, Guo CF. Flexible electronics: Stretchable electrodes and their future. *Adv. Funct. Mater.* **29**, 1805924 (2019).
14. Gutruf P, Walia S, Nur Ali M, Sriram S, Bhaskaran M. Strain response of stretchable micro-electrodes: Controlling sensitivity with serpentine designs and encapsulation. *Appl. Phys. Lett.* **104**, 021908 (2014).
15. Lacour SP, Chan D, Wagner S, Li T, Suo Z. Mechanisms of reversible stretchability of thin metal films on elastomeric substrates. *Appl. Phys. Lett.* **88**, 204103 (2006).
16. Liu Z, *et al.* Surface Strain Redistribution on Structured Microfibers to Enhance Sensitivity of Fiber-Shaped Stretchable Strain Sensors. *Adv. Mater.* **30**, 1704229 (2018).
17. Deng J, *et al.* Electrical bioadhesive interface for bioelectronics. *Nat. Mater.* **20**, 229-236 (2021).
18. Hwang H, *et al.* Stretchable anisotropic conductive film (S-ACF) for electrical interfacing in high-resolution stretchable circuits. *Sci. Adv.* **7**, eabh0171 (2021).

19. Inoue A, Yuk H, Lu B, Zhao X. Strong adhesion of wet conducting polymers on diverse substrates. *Sci. Adv.* **6**, eaay5394.
20. You I, *et al.* Artificial multimodal receptors based on ion relaxation dynamics. *Science* **370**, 961-965 (2020).
21. Yuan X-Z, Song C, Wang H, Zhang J. *Electrochemical impedance spectroscopy in PEM fuel cells: fundamentals and applications*. Springer (2010).
22. Andrew KJ. Dielectric relaxation in solids. *J. Phys. D: Appl. Phys.* **32**, R57 (1999).
23. Gerhardt R. Impedance and dielectric spectroscopy revisited: Distinguishing localized relaxation from long-range conductivity. *J. Phys. Chem. Solids* **55**, 1491-1506 (1994).

REVIEWER COMMENTS

Reviewer #1 (Remarks to the Author):

Comments addressed

Reviewer #2 (Remarks to the Author):

The authors provide detailed responses including in-depth analysis, comparisons with other studies, and additional demonstrations to support their claims. They discuss the superior interfacial binding strength, electrical properties, and anti-friction abilities of their Au-RPU-based devices, comparing them with other materials and demonstrating their practical applications.

The response seems thorough, aiming to address the reviewer's concerns by providing comprehensive technical details and experimental results.

Their detailed approach in addressing the reviewer's concerns through technical and experimental evidence supports the manuscript's suitability for publication in Nature Communications in its present form.

Reviewer #3 (Remarks to the Author):

The authors have clearly addressed to our comments and suggested a kind of supporting data to make sure their originality.

In this regard, I recommend the revised manuscript to be published in the Nature Communications Journal without additional revision.

Detailed responses to the referees' comments

Reviewer #1:

Comments: *Comments addressed.*

Response: Thank you for your previous valuable comments on our work.

Reviewer #2:

Comments: *The authors provide detailed responses including in-depth analysis, comparisons with other studies, and additional demonstrations to support their claims. They discuss the superior interfacial binding strength, electrical properties, and anti-friction abilities of their Au-RPU-based devices, comparing them with other materials and demonstrating their practical applications.*

The response seems thorough, aiming to address the reviewer's concerns by providing comprehensive technical details and experimental results.

Their detailed approach in addressing the reviewer's concerns through technical and experimental evidence supports the manuscript's suitability for publication in Nature Communications in its present form.

Response: Thank you for your excellent contribution to our manuscript, which improves the quality of this work significantly.

Reviewer #3:

Comments: *The authors have clearly addressed to our comments and suggested a kind of supporting data to make sure their originality.*

In this regard, I recommend the revised manuscript to be published in the Nature Communications Journal without additional revision.

Response: Thank you very much.